The EMBO Journal (2013) 32, 2231–2247
www.embojournal.org

# A direct physical interaction between Nanog and Sox2 regulates embryonic stem cell self-renewal

Alessia Gagliardi[1,5], Nicholas P Mullin[1,5], Zi Ying Tan[1,6], Douglas Colby[1], Anastasia I Kousa[1], Florian Halbritter[1], Jason T Weiss[1,7], Anastasia Felker[1,8], Karel Bezstarosti[2], Rebecca Favaro[3], Jeroen Demmers[2], Silvia K Nicolis[3], Simon R Tomlinson[1], Raymond A Poot[4] and Ian Chambers[1,*]

[1]MRC Centre for Regenerative Medicine, Institute for Stem Cell Research, School of Biological Sciences, University of Edinburgh, Edinburgh, UK, [2]Proteomics Center, Erasmus MC, Rotterdam, The Netherlands, [3]Department of Biotechnology and Biological Sciences, University of Milano-Bicocca, Milano, Italy and [4]Department of Cell Biology, Erasmus MC, Rotterdam, The Netherlands

**Embryonic stem (ES) cell self-renewal efficiency is determined by the Nanog protein level. However, the protein partners of Nanog that function to direct self-renewal are unclear. Here, we identify a Nanog interactome of over 130 proteins including transcription factors, chromatin modifying complexes, phosphorylation and ubiquitination enzymes, basal transcriptional machinery members, and RNA processing factors. Sox2 was identified as a robust interacting partner of Nanog. The purified Nanog–Sox2 complex identified a DNA recognition sequence present in multiple overlapping Nanog/Sox2 ChIP-Seq data sets. The Nanog tryptophan repeat region is necessary and sufficient for interaction with Sox2, with tryptophan residues required. In Sox2, tyrosine to alanine mutations within a triple-repeat motif (S X T/S Y) abrogates the Nanog–Sox2 interaction, alters expression of genes associated with the Nanog-Sox2 cognate sequence, and reduces the ability of Sox2 to rescue ES cell differentiation induced by endogenous Sox2 deletion. Substitution of the tyrosines with phenylalanine rescues both the Sox2–Nanog interaction and efficient self-renewal. These results suggest that aromatic stacking of Nanog tryptophans and Sox2 tyrosines mediates an interaction central to ES cell self-renewal.**

*The EMBO Journal* (2013) **32**, 2231–2247. doi:10.1038/emboj.2013.161; Published online 26 July 2013
Subject Categories: signal transduction; development

*Corresponding author. MRC Centre for Regenerative Medicine, Institute for Stem Cell Research, School of Biological Sciences, University of Edinburgh, 5 Little France Drive, Edinburgh EH16 4UU, UK.
Tel.: +44 (0)131 651 9500; Fax: +44 (0)131 651 9501;
E-mail: ichambers@ed.ac.uk
[5]These authors contributed equally to this work.
[6]Present address: Gene Regulation Laboratory, Genome Institute of Singapore, Singapore 138672, Singapore.
[7]Present address: Edinburgh Cancer Research UK Centre, MRC Institute of Genetics and Molecular Medicine, University of Edinburgh, Crewe Road South, Edinburgh EH4 2XR, UK.
[8]Present address: Institute of Molecular Life Sciences, University of Zurich, Winterthurerstrasse 190, 8057 Zurich, Switzerland.

Keywords: DNA-independent interaction; hydrophobic stacking; pluripotency; protein interactome; SELEX

## Introduction

Embryonic stem (ES) cell self-renewal efficiency depends on the level of expression of components of the pluripotency gene regulatory network. Among these, Oct4, Sox2 and Nanog play central roles. While the levels of Oct4 and Sox2 are relatively uniform in undifferentiated ES cells, the levels of Nanog vary considerably (Hatano *et al*, 2005; Chambers *et al*, 2007; Singh *et al*, 2007) with high levels of Nanog directing efficient self-renewal (Chambers *et al*, 2003, 2007). However, the mechanisms by which Nanog delivers this function in ES cells are not fully understood. In particular, although Nanog has been reported to interact with several proteins (Wang *et al*, 2006; Wu *et al*, 2006; Liang *et al*, 2008; Costa *et al*, 2013), the full extent of the Nanog interactome is not known.

In the past few years, proteomic approaches have been employed to characterize and begin to understand the network of biochemical interactions controlling pluripotent cell function. This has resulted in the identification of additional proteins that interact with the key transcriptional factors Nanog, Sox2 and Oct4 to control and maintain the pluripotent state. Pioneering studies by Wang *et al* (2006) identified a Nanog-centred interactome of 17 proteins that extended to other transcription factors including Oct4, Zfp281, Nac1, Rex1 and Nr0b1. This list of Nanog-interacting proteins has since been extended (Liang *et al*, 2008) with a recent interactome identifying a total of 27 Nanog interactors (Costa *et al*, 2013). This relatively small number is in contrast to the larger number of interactors identified in recent Oct4 (Pardo *et al*, 2010; van den Berg *et al*, 2010; Ding *et al*, 2012) and Sox2 (Gao *et al*, 2012) interactomes. Interactome studies have the potential to contribute to the elucidation of the mechanisms by which specific factors function. Central to this is the identification of the interacting amino-acid side chains on partner proteins as well as the functional significance of their association. To date, biochemical characterization of protein–protein interactions in pluripotent cells has been relatively sparse with most effort analysing the interaction between Sox2 and Oct4 (Yuan *et al*, 1995; Ambrosetti *et al*, 1997, 2000; Remenyi *et al*, 2003; Kim *et al*, 2008; Chen *et al*, 2008a; Lam *et al*, 2012). From a biochemical perspective, little is known about how Nanog fits into the tight relationship between Oct4 and Sox2.

Previously, we described a method to identify partner proteins interacting with nuclear proteins of interest in ES cells and used this to identify an extensive interaction network for the transcription factor Oct4 (van den Berg *et al*, 2010). Here, this technique is applied to Nanog, resulting in identification of a Nanog interactome which includes over 130 Nanog partners in ES cells. From this, the direct

interaction between Nanog and Sox2 was selected for further characterization, pinpointing individual residues required for the interaction and defining the functional consequences of elimination of the interaction between these central pluripotency regulators.

## Results

### Identification of a Nanog interactome

An ES cell line expressing epitope-tagged Nanog protein was obtained by transfection of E14Tg2a cells with a construct in which the constitutive CAG promoter directs expression of (FLAG)$_3$Nanog, linked via an IRES to puromycin resistance. Puromycin-resistant colonies were expanded and the resulting cell lines were analysed by immunoblotting. A cell line was identified (hereafter called F-Nanog) that expressed (FLAG)$_3$Nanog at close to endogenous levels (Figure 1A). A qRT-PCR analysis of F-Nanog and E14Tg2a wild-type cells showed no significant difference in the expression level of the ES cell-specific genes Oct4, Sox2 and Rex1 (Figure 1B). In agreement with recent reports of autorepression by Nanog (Fidalgo *et al*, 2012; Navarro *et al*, 2012b), F-Nanog cells

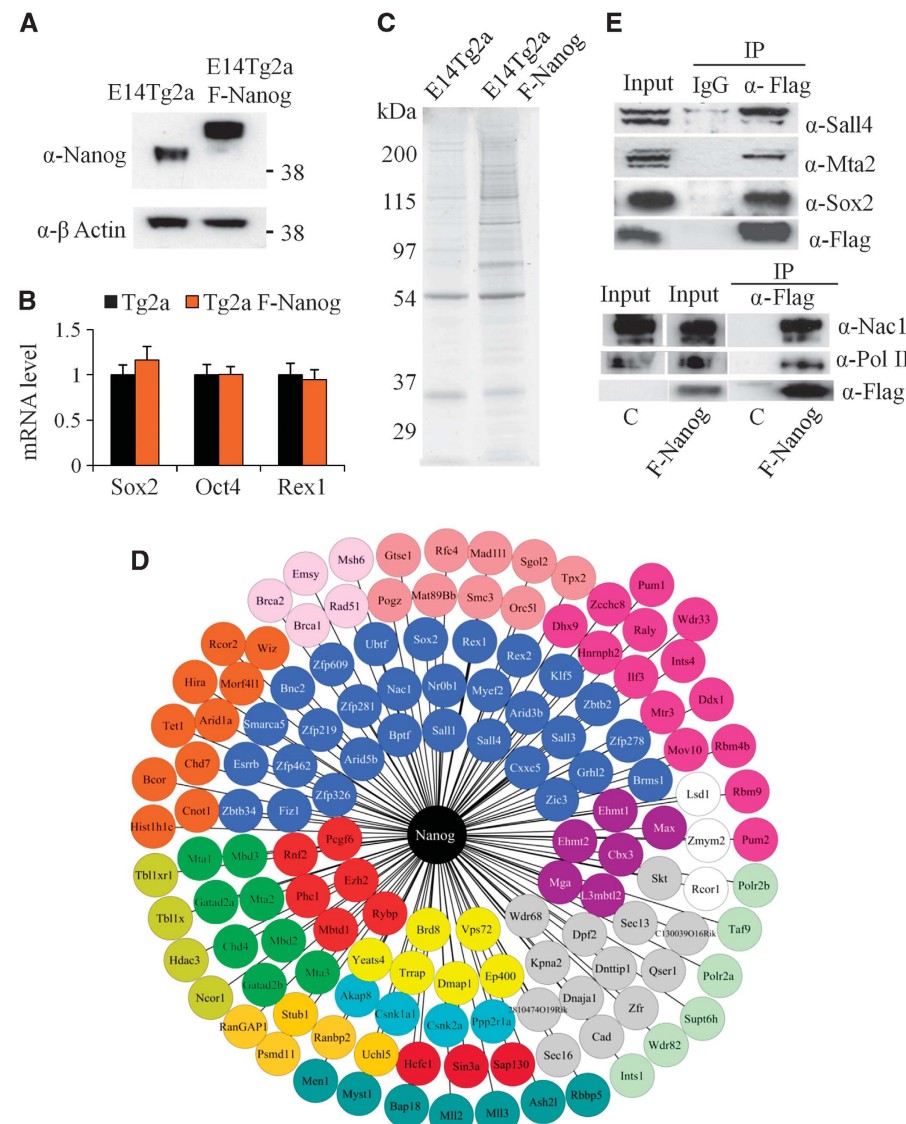

**Figure 1** Characterization of E14Tg2a Flag Nanog cell line. (**A**) Expression levels of Nanog protein in E14Tg2a and E14Tg2a F-Nanog cells compared by immunoblot analysis using β-actin as a loading control. Note the reduced expression of endogenous Nanog protein in cells transfected with (Flag)$_3$Nanog, consistent with autorepression of the *Nanog* gene by Nanog protein (Navarro *et al*, 2012a). (**B**) Expression levels of Sox2, Oct4 and Rex1 in E14Tg2a F-Nanog relative to E14Tg2a which was set to 1. Error bars are s.e.m. of three biological replicates. (**C**) Coomassie-stained SDS–polyacrylamide gel of the FLAG immunoprecipitation from E14Tg2a F-Nanog and control E14Tg2a cells. (**D**) Proteins detected by mass spectrometry analysis are grouped in classes. Transcription factors are shown in blue circles, NuRD components are in green, Trrap/p400 complex is in yellow, PcG components are in red, E2F6 complex is in purple, Sin3a complex is in burgundy, N-CoR complex is in khaki, LSD1 complex is white, Mll complex is in blue green, chromatin remodelling/transcriptional regulation proteins are in dark orange, transcriptional machinery proteins are in pale green, proteins involved in phosphorylation are in pale blue, proteins involved in ubiquitination are in amber, proteins involved in RNA processing are in fuschia, proteins involved in cell cycle or DNA replication are in coral, proteins involved in DNA repair are in pink and other proteins are in grey. (**E**) Nuclear extracts from E14Tg2a F-Nanog cells (top) or from RCNβH-B(t):F-Nanog (bottom) were immunoprecipitated as indicated and immunoblots analysed with the indicated antibodies. In the bottom panel, C refers to control samples from RCNβH-B(t) parental cells. Source data for this figure is available on the online supplementary information page.

show a strong decrease in expression of endogenous Nanog, which has the fortuitous consequence of maximizing the proportion of Nanog protein immunoprecipitated by anti-FLAG reagents.

Nuclear extracts were prepared from F-Nanog cells and parental E14Tg2a cells and used for FLAG-affinity purifications as previously described (van den Berg *et al*, 2010). A Coomassie-stained gel of the eluted fractions showed several bands absent from the control E14Tg2a sample, indicating good signal-to-background ratio (Figure 1C). Mass spectrometry analysis was then performed on two independent affinity purifications from both F-Nanog and E14Tg2a control cells. An extensive set of Nanog partners was identified that could be grouped into several functional categories (Table I; Supplementary Table I; Figure 1D). The group with the highest representation is transcription factors, other groups present being chromatin modification complexes (e.g., NuRD and NcoR), proteins involved in phosphorylation or ubiquitination, basal transcriptional machinery members and RNA processing proteins. Mass spectrometric analysis of an independent cell line generated by expressing the same (Flag)$_3$Nanog expression cassette in a Nanog-null ES cell line (RCNβH-B(t)) (Chambers *et al*, 2007) was used to verify candidate Nanog-interacting proteins (Table I; Supplementary Table I). Only the proteins identified in two out of three purifications are included in Table I and Supplementary Table I. Interactions between Nanog and Sox2, RNA polymerase II (RNAPolII), Nac-1, Sall4 and the NuRD subunit Mta2 were also observed by immunoblotting (Figure 1E). The Nanog interactome substantially overlaps with the published interactome of Oct4, Esrrb, Sall4, Nr0b1 and Tcfcp2l1 (van den Berg *et al*, 2010) (Figure 2). Interestingly, Chd7 and the Ncor1 complex, which are not part of the Oct4/Esrrb/Sall4/Nr0b1/Tcfcp2l1 interactome, do interact with Nanog (Figure 2). This may reflect the robust interaction of Nanog with Sox2 (Table I; Supplementary Table I) as both Chd7 and the Ncor1 complex interact with Sox2 (Engelen *et al*, 2011).

### Mapping the domain of Nanog interacting with Sox2

Due to the key role of Sox2 in ES cell biology, further characterization of the interaction between Nanog and Sox2 was undertaken. To determine whether the interaction between Nanog and Sox2 could be detected in wild-type ES cells, E14Tg2a nuclear extract was incubated either with an anti-Sox2 antibody and immunoprecipitates examined for the presence of Nanog or with an anti-Nanog antibody and immunoprecipitates examined for Sox2. Nanog was detected in Sox2 immunoprecipitates (Figure 3A) and Sox2 was also detected in Nanog immunoprecipitates (Figure 3B). To map the sites of interaction, co-transfections of (Flag)$_3$Sox2 with (HA)$_3$Nanog or Nanog deletion mutants were performed (Figure 3C) in E14/T cells (Chambers *et al*, 2003). Nuclear extracts from ES cells transfected with (HA)$_3$Nanog and (HA)$_3$-tagged mutants lacking the N-terminus, the DNA binding homeodomain (HD) or the C-terminus of Nanog were immunoprecipitated with the HA antibody and after SDS–PAGE, immunoblots were probed for the presence of interacting Sox2 using a Flag antibody. (Flag)$_3$Sox2 does not interact with a Nanog mutant lacking the C-terminal domain but the interaction between Sox2 and Nanog variants lacking either the N-terminus or the HD remained intact (Figure 3C).

To identify the subregion of the Nanog C-terminal domain responsible for the interaction with Sox2, (Flag)$_3$Sox2 was co-transfected with (HA)$_3$Nanog variants carrying mutations within the C-terminal domain. Co-immunoprecipitations showed that deletion of the tryptophan repeat (WR) region, within the C-terminal domain of Nanog, but not residues C-terminal to the WR, abrogated the interaction with Sox2 (Figure 3D). Importantly, a Nanog mutant in which all 10 tryptophan residues in the WR region were mutated to alanines, (HA)$_3$Nanog WR$_{W10-A}$, also failed to interact with Sox2, pinpointing the tryptophan residues as critical determinants of the interaction with Sox2. To determine whether the interaction of Nanog and Sox2 was direct, Sox2 was co-expressed in *E. coli* alongside a fusion between Maltose Binding Protein and, either the Nanog tryptophan repeat, or the Nanog tryptophan repeat in which all the tryptophans were replaced by alanines (MBP-WR or MBP-WR$_{W10-A}$) (Figure 3E). The MBP-fusion proteins were then purified on an amylose column and any interacting Sox2 was detected by immunoblotting with a Sox2 antibody. Only MBP-WR but not MBP-WR$_{W10-A}$ was able to co-precipitate Sox2 (Figure 3E). Taken together, these experiments indicate that Nanog and Sox2 interact directly, that the interaction with Sox2 can be mediated by the Nanog WR domain alone and that tryptophan residues within the WR are required for interaction with Sox2. In addition, the ability of these proteins to interact in *E. coli* implies that post-translational modifications are not required for interaction between Nanog and Sox2.

### The region of Sox2 interacting with Nanog

To identify the region of Sox2 involved in the interaction with Nanog, we investigated mutants carrying deletions within the C-terminal domain, the HMG DNA binding domain or residues at the N-terminus of Sox2 (Figure 4A). Each of these mutants was co-expressed with (HA)$_3$Nanog in E14/T cells, nuclear extracts prepared and the HA antibody used to co-immunoprecipitate (HA)$_3$Nanog and interacting proteins. Samples were then analysed by SDS–PAGE and immunoblotting. (Flag)$_3$Sox2 mutants lacking the N-terminal region, the DNA binding domain or the C-terminal 56 amino acid residues [(Flag)$_3$Sox2 1-263] were still able to interact with Nanog (Figure 4A). However, (Flag)$_3$Sox2 1-204 does not interact with Nanog, suggesting that the serine-rich region is involved in the interaction with the Nanog WR.

The persistence of the Nanog–Sox2 interaction in nuclear extracts that have been treated with the nuclease, benzonase, to eliminate interactions mediated via DNA bridging, suggests that DNA binding is not required for the Nanog–Sox2 interaction. Moreover, the above results indicate that Nanog and Sox2 can interact in the absence of a DNA binding domain on either of the proteins (Figures 3C and 4A). To consolidate the notion that Nanog–Sox2 interaction is fully DNA independent, we show by co-immunoprecipitation of (Flag)$_3$Sox2ΔHMG and (HA)$_3$NanogΔHD that Nanog and Sox2 molecules that lack the DNA binding domains can still interact (Figure 4B).

Our analysis of the ability of (HA)$_3$Nanog to co-immunoprecipitate Sox2 mutants (Figure 4) suggested that the serine-rich region, from residues 205 to 263, plays a key role in the Nanog interaction. To narrow down the region of Sox2 interacting with Nanog, further deletion mutants within this

**Table I** Nanog-interacting proteins as identified by mass spectrometry analysis of purified Nanog samples

| Protein | Accession | E14Tg2a F-Nanog #1 | | E14Tg2a F-Nanog #2 | | RCNβH-B(t) F-Nanog #1 | | |
|---|---|---|---|---|---|---|---|---|
| | | Mascot[a] | Peptides[b] | Mascot[a] | Peptides[b] | Mascot[a] | Peptides[b] | Average Mascot |
| Nanog | Q80Z64 | 445 | 7 | 339 | 5 | 406 | 7 | 397 |
| *Transcription factors* | | | | | | | | |
| Sall4 | Q8BX22 | 1880 (425) | 26 (8) | 1818 (391) | 28 (9) | 2398 (918) | 35 (19) | 2032 |
| Bptf | A2A654 | 2402 | 38 | 1804 | 42 | 362 | 9 | 1523 |
| Sall1 | Q6P5E3 | 1306 | 20 | 1447 | 25 | 1696 (185) | 30 (3) | 1483 |
| Smarca5 | Q91ZW3 | 1453 (721) | 25 (15) | 1099 (364) | 26 (12) | 506 | 11 | 1019 |
| Esrrb | E9QKA2 | 841 (88) | 13 (2) | 663 | 12 | 860 (52) | 15 (1) | 788 |
| Zfp462 | B1AWL2 | 811 | 18 | 470 | 11 | 1067 | 26 | 783 |
| Zfp281 | Q99LI5 | 753 | 12 | 751 | 16 | 759 | 14 | 754 |
| Nr0b1 | Q61066 | 729 | 9 | 761 | 12 | 647 | 12 | 712 |
| Myef2 | Q99N20 | 519 (424) | 9 (8) | 726 (217) | 12 (4) | 726 (133) | 14 (3) | 657 |
| Arid3b | Q99LI5 | 490 (85) | 6 (1) | 719 | 12 | 435 | 9 | 548 |
| Nac1 | Q7TSZ8 | 601 (75) | 10 (1) | 610 (80) | 13 (2) | 898 | 15 | 527 |
| Zfp219 | Q6IQX8 | 637 | 11 | 580 | 11 | 548 | 11 | 441 |
| Arid5b | Q8BM75 | 547 | 10 | 622 | 13 | 117 | 2 | 429 |
| Sall3 | Q62255 | 336 | 5 | 312 | 5 | 583 | 11 | 410 |
| Bnc2 | E9Q5T4 | 271 | 5 | 354 | 9 | 431 | 9 | 352 |
| Zfp609 | E9Q1Y2 | 380 | 7 | 467 | 9 | 511 | 10 | 340 |
| Ubtf | P25976 | 566 (101) | 12 (2) | 223 | 5 | 60 | 2 | 283 |
| Sox2 | Q60I23 | 264 | 5 | 250 | 4 | 307 | 5 | 274 |
| Rex1 | P22227 | 209 (47) | 3 (1) | 65 | 1 | 541 (130) | 9 (3) | 272 |
| Rex2 | Q6NWW4 | 303 (66) | 4 (2) | 208 | 4 | 199 | 4 | 237 |
| Klf5 | Q9Z0Z7 | 168 | 3 | 88 | 2 | 418 (71) | 9 (2) | 225 |
| Zbtb2 | Q3V3W4 | 125 | 2 | 200 | 4 | 275 (52) | 5 (1) | 200 |
| Cxxc5 | Q91WA4 | 148 | 3 | 216 | 4 | 213 | 5 | 192 |
| Zfp326 | O88291 | 330 | 5 | 366 | 7 | 64 | 1 | 190 |
| Fiz1 | Q9WTJ4 | 233 (74) | 4 (2) | 125 (79) | 2 (2) | 201 | 3 | 186 |
| Zfp278 | Q80XS2 | 204 | 4 | 272 | 8 | 75 | 2 | 184 |
| Zbtb34 | A2ATY4 | 118 | 1 | 42 | 1 | 227 | 4 | 173 |
| Zic3 | Q62521 | 249 | 4 | 116 | 3 | 105 | 2 | 157 |
| Brms1 | Q99N20 | 221 | 4 | 62 | 1 | 118 | 3 | 134 |
| Grhl2 | Q8K5C0 | 88 | 2 | 127 | 3 | — | — | 72 |
| *NuRD complex* | | | | | | | | |
| Chd4 | Q6PDQ2 | 2847 (560) | 55 (13) | 1917 (160) | 44 (5) | 942 | 22 | 1902 |
| Gatad2a | Q8CHY6 | 938 (50) | 17 (1) | 1331 (128) | 21 (3) | 1566 (244) | 25 (5) | 1278 |
| Mta1 | F8WHY8 | 988 (272) | 17 (4) | 1220 (355) | 17 (8) | 726 | 14 | 978 |
| Gatad2b | Q8VHR5 | 692 (170) | 10 (3) | 950 (141) | 17 (3) | 1183 (472) | 19 (9) | 942 |
| Mta2 | Q9R190 | 775 (154) | 12 (5) | 1201 (150) | 22 (4) | 796 | 17 | 924 |
| Mta3 | E9Q794 | 715 (245) | 12 (4) | 820 | 16 | 675 | 12 | 737 |
| Mbd3 | D3YTR4 | 746 (155) | 12 (3) | 469 (96) | 7 (3) | 777 (264) | 13 (5) | 664 |
| Mbd2 | Q9Z2E1 | 50 | 2 | 95 | 3 | 152 | 4 | 99 |
| *Trrap/p400 complex* | | | | | | | | |
| Trrap | E9QLK7 | 2434 (565) | 48 (14) | 1067 (84) | 26 (2) | 1241 | 29 | 1581 |
| Ep400 | Q8CHI8 | 898 (45) | 17 (2) | 483 | 14 | 695 | 15 | 797 |
| Yeats4 | Q9CR11 | 435 | 9 | 261 | 6 | 114 | 3 | 270 |
| Dmap1 | Q9JI44 | 224 (114) | 2 (1) | 328 (68) | 7 (2) | 235 | 4 | 262 |
| Vps72 | Q62481 | 142 | 3 | 73 | 2 | 116 | 3 | 110 |
| Brd8 | D3YZC7 | 117 | 2 | — | — | 192 | 3 | 103 |
| *Polycomb complex* | | | | | | | | |
| Mbtd1 | Q6P5G3 | 136 | 3 | 166 | 3 | 197 | 3 | 166 |
| Phc1 | Q64028 | 537 | 10 | 486 | 9 | 415 | 8 | 479 |
| Rnf2 | Q9CQJ4 | 446 | 9 | 311 | 7 | 312 (155) | 5 (3) | 356 |
| Rybp | Q8CCI5 | 277 (395) | 4 (5) | 163 | 3 | 383 (54) | 6 (1) | 274 |
| Ezh2 | E9QNF8 | 94 | 1 | — | — | 212 | 5 | 102 |
| Pcgf6 | Q99NA9 | 115 | 2 | — | — | 163 | 4 | 93 |
| *E2F6 complex* | | | | | | | | |
| Mga | E9QLG3 | 817 | 13 | 752 | 18 | 395 | 9 | 655 |
| Cbx3 | P23198 | 499 (367) | 6 (5) | 282 (75) | 4 (2) | 358 (83) | 6 (2) | 380 |
| Rnf2 | Q9CQJ4 | 446 | 9 | 311 | 7 | 312 (155) | 5 (3) | 356 |
| Ehmt1 | Q5DW34 | 229 | 5 | 111 | 3 | 429 | 9 | 256 |
| Ehmt2 | Q9Z148 | — | — | 57 | 2 | 491 | 12 | 183 |
| L3mbtl2 | P59178 | 161 (47) | 3 (1) | 242 (41) | 6 (1) | 101 | 1 | 168 |
| Max | P28574 | 122 | 2 | 158 | 3 | — | — | 93 |

**Table I** Continued

| | | E14Tg2a F-Nanog #1 | | E14Tg2a F-Nanog #2 | | RCNβH-B(t) F-Nanog #1 | | |
|---|---|---|---|---|---|---|---|---|
| *Sin3a complex* | | | | | | | | |
| Sin3a | Q60520 | 1123 (154) | 19 (3) | 765 (143) | 16 (5) | 1221 (223) | 22 (6) | 1036 |
| Hcfc1 | Q61191 | 546 (291) | 9 (7) | 419 (114) | 12 (3) | 926 | 20 | 630 |
| Ncor1 | E9Q2B2 | 237 | 4 | 358 | 7 | 254 | 5 | 283 |
| Sap130 | Q8BIH0 | 144 | 3 | 69 | 2 | 220 | 4 | 144 |
| Mbd2 | Q9Z2E1 | 50 | 2 | 95 | 3 | 152 | 4 | 99 |
| | | | | | | | | |
| *N-CoR complex* | | | | | | | | |
| Tbl1xr1 | Q8BHJ5 | 962 (289) | 13 (5) | 837 (236) | 13 (4) | 292 | 5 | 697 |
| Tbl1x | Q9QXE7 | 677 | 10 | 663 | 12 | 479 | 8 | 606 |
| Ncor1 | E9Q2B2 | 237 | 4 | 358 | 7 | 254 | 5 | 283 |
| Hdac3 | Q3UM33 | 226 | 6 | 250 | 5 | 69 | 2 | 182 |
| | | | | | | | | |
| *LSD1 complex* | | | | | | | | |
| Lsd1 | Q6ZQ88 | 589 (237) | 12 (4) | 647 (106) | 12 (3) | 924 (58) | 17 (2) | 720 |
| Zmym2 | Q9CU65 | 155 | 2 | 281 | 6 | 1005 | 21 | 480 |
| Rcor1 | E9QK09 | 68 | 1 | — | — | 91 | 2 | 53 |
| | | | | | | | | |
| *MLL complex* | | | | | | | | |
| Hcfc1 | Q61191 | 546 (291) | 9 (7) | 419 (114) | 12 (3) | 926 | 20 | 630 |
| Bap18 | Q9DCT6 | 663 (47) | 14 (1) | 933 | 19 | 222 | 4 | 606 |
| Mll2 | O08550 | 668 | 16 | 195 | 7 | 133 | 3 | 332 |
| Mll3 | Q8BRH4 | 670 | 14 | 259 | 8 | — | — | 310 |
| Ash2l | E9PU93 | 174 | 4 | 340 | 7 | 273 | 6 | 204 |
| Men1 | O88559 | 68 | 2 | 125 | 3 | 245 | 4 | 146 |
| Myst1 | Q9D1P2 | 131 | 3 | 100 | 3 | 205 | 4 | 145 |
| Rbbp5 | Q8BX09 | 114 | 3 | 219 | 6 | 76 | 2 | 63 |
| | | | | | | | | |
| *Chromatin remodelling/transcriptional regulation* | | | | | | | | |
| Chd7 | A2AJK6 | 1408 (154) | 24 (4) | 944 | 17 | 252 | 7 | 868 |
| Cnot1 | Q6ZQ08 | 1471 (355) | 33 (9) | 710 (221) | 18 (6) | 245 (62) | 6 (2) | 809 |
| Tet 1 | E9Q9Y4 | 1370 | 24 | — | — | 390 | 9 | 587 |
| Arid1a | E9QAQ7 | 83 | 3 | 182 | 5 | 1315 (179) | 27 (4) | 527 |
| Rcor2 | Q8C796 | 240 (99) | 4 (2) | 423 (48) | 7 (1) | 694 | 12 | 452 |
| Morf4l1 | P60762 | 458 (47) | 8 (1) | 446 | 10 | 367 | 7 | 424 |
| Wiz | F6ZBR8 | 247 | 4 | 185 | 4 | 769 | 17 | 400 |
| Bcor | Q8CGN4 | 322 | 7 | 454 | 11 | 279 | 6 | 352 |
| Hira | Q61666 | 451 | 7 | 394 | 11 | 102 | 3 | 316 |
| Hist1h1c | P15864 | 274 | 4 | 139 | 3 | 112 | 2 | 175 |
| | | | | | | | | |
| *Transcriptional machinery* | | | | | | | | |
| Ints1 | Q6P4S8 | 721 (172) | 14 (5) | 608 (213) | 13 (6) | 248 | 6 | 526 |
| Wdr82 | Q8BFQ4 | 592 (431) | 9 (8) | 320 (69) | 8 (2) | 377 (78) | 8 (1) | 430 |
| Supt6h | Q62383 | 868 | 18 | 252 | 7 | 101 | 3 | 407 |
| Polr2a | P08775 | 240 (93) | 2 | 148 | 4 | 481 | 11 | 290 |
| Taf9 | Q8VI33 | 317 | 7 | 158 | 4 | 261 | 7 | 245 |
| Polr2b | Q8CFI7 | 281 | 5 | 122 | 3 | — | — | 134 |
| | | | | | | | | |
| *Phosphorylation* | | | | | | | | |
| Akap8 | Q9DBR0 | 827 (190) | 14 (3) | 760 (162) | 13 (3) | 859 (75) | 15 (1) | 815 |
| Csnk2a | Q60737 | 408 (118) | 8 (3) | 323 (45) | 7 (2) | 431 | 8 | 387 |
| Csnk1a1 | E9Q2U6 | 247 | 5 | 202 | 5 | 151 | 4 | 200 |
| Ppp2r1a | Q76MZ3 | 178 (67) | 4 (2) | 261 | 6 | 57 | 2 | 165 |
| | | | | | | | | |
| *Ubiquitination* | | | | | | | | |
| Uchl5 | Q9WUP7 | 349 (101) | 6 (2) | 344 (96) | 8 (2) | 238 (88) | 5 (2) | 310 |
| Stub1 | Q9WUD1 | 457 | 10 | 350 | 7 | 83 | 2 | 297 |
| Ranbp2 | Q9ERU9 | 283 (90) | 8 (2) | 164 (44) | 4 (1) | 932 (42) | 20 (3) | 220 |
| RanGAP1 | E9Q757 | 118 (47) | 3 (1) | 289 | 6 | 370 (85) | 7 (2) | 164 |
| Psmd11 | Q8BG32 | 217 (70) | 6 (2) | 137 (43) | 4 (2) | — | — | 118 |
| | | | | | | | | |
| *RNA processing* | | | | | | | | |
| Hnrnph2 | P70333 | 513 | 8 | 625 (174) | 11 (5) | 545 | 10 | 2437 |
| Dhx9 | O70133 | 972 | 17 | 502 | 12 | 1126 (530) | 22 (13) | 867 |
| Ilf3 | Q45VK5 | 1388 (689) | 22 (14) | 1022 (230) | 21 (7) | 670 | 13 | 803 |
| Raly | Q3U3F6 | 480 (488) | 9 (10) | 392 (119) | 7 (3) | 271 | 7 | 381 |
| Zcchc8 | Q9CYA6 | 425 | 5 | 377 | 6 | 274 | 5 | 359 |
| Ddx1 | Q91VR5 | 167 (354) | 4 (8) | 333 (63) | 7 (2) | 259 | 6 | 253 |
| Wdr33 | Q8K4P0 | 316 (42) | 6 (1) | 221 (77) | 6 (3) | 219 | 6 | 252 |
| Pum1 | Q80U78 | 274 (120) | 6 (3) | 290 (66) | 7 (2) | 177 | 4 | 247 |
| Rbm9 | Q8BP71 | 214 | 4 | 305 | 4 | 210 | 4 | 243 |

**Table I** Continued

| | | E14Tg2a F-Nanog #1 | | E14Tg2a F-Nanog #2 | | RCNβH-B(t) F-Nanog #1 | | |
|---|---|---|---|---|---|---|---|---|
| Mtr3 | Q8BTW3 | 380 (74) | 5 (1) | 120 (108) | 2 (2) | 212 | 4 | 237 |
| Mov10 | P23249 | 251 (45) | 6 (1) | 185 (58) | 3 (2) | — | — | 145 |
| Rbm4b | Q8VE92 | 87 | 2 | 61 | 2 | 279 | 6 | 142 |
| Ints4 | Q8CIM8 | 112 | 3 | 122 | 3 | 192 | 4 | 142 |
| Pum2 | Q80U58 | 122 | 2 | 187 | 4 | — | — | 103 |
| *Cell cycle/DNA replication* | | | | | | | | |
| Sgol2 | Q7TSY8 | 747 (91) | 12 (3) | 387 (52) | 8 (2) | 735 | 13 | 623 |
| Mad1l1 | Q9WTX8 | 493 (126) | 9 (3) | 520 | 11 | 732 (984) | 14 (20) | 582 |
| Rfc4 | Q99J62 | 517 (171) | 13 (5) | 541 (265) | 10 (7) | 324 (78) | 6 (2) | 461 |
| Gtse1 | Q8R080 | 635 (178) | 9 (3) | 587 | 11 | 64 | 1 | 429 |
| Tpx2 | A2APB8 | 374 (40) | 8 (1) | 142 | 3 | 383 | 9 | 300 |
| Orc5l | Q9WUV0 | 196 (45) | 4 (1) | 109 (140) | 3 (4) | 260 | 5 | 188 |
| Smc3 | Q9CW03 | 124 | 3 | — | — | 338 | 9 | 154 |
| Mat89Bb | Q8QZV7 | 125 (30) | 3 (1) | 144 | 5 | 184 | 4 | 151 |
| Pogz | D3YUW8 | 91 | 1 | 40 | 1 | 268 | 6 | 180 |
| *DNA repair* | | | | | | | | |
| Msh6 | P54276 | 613 (112) | 15 (3) | 623 (139) | 14 (4) | 208 | 6 | 481 |
| Brca2 | P97929 | 576 | 14 | 309 | 9 | 296 | 9 | 394 |
| Rad51 | D6RCK1 | 328 | 5 | 208 | 4 | 232 | 5 | 256 |
| Brca1 | A2A4Q4 | 74 | 2 | 157 | 4 | 466 | 12 | 239 |
| Emsy | Q8BMB0 | 306 (52) | 6 (2) | 149 | 5 | 69 | 2 | 152 |
| *Other* | | | | | | | | |
| Sec16 | E9QAT4 | 889 (44) | 15 (1) | 777 | 18 | 353 | 8 | 673 |
| Cad | O54788 | 883 | 18 | 968 | 24 | 108 | 1 | 653 |
| Qser1 | A2BIE1 | 669 (192) | 11 (4) | 433 | 9 | 826 | 16 | 643 |
| Zfr | O88532 | 241 (80) | 4 (1) | 193 (67) | 4 (1) | 1155 | 21 | 530 |
| Dnaja1 | P63037 | 515 (144) | 9 (2) | 345 (107) | 8 (2) | 514 | 8 | 458 |
| 2810474O19RIK | D3Z687 | 633 (47) | 14 (1) | 933 | 19 | — | — | 380 |
| C130039O16Rik | E9Q2I4 | 474 (41) | 7 (1) | 314 | 7 | 258 | 5 | 349 |
| Skt | A2AQ25 | 247 | 5 | 665 | 15 | 60 | 2 | 324 |
| Wdr68 | P61963 | 385 | 5 | 381 | 8 | 168 | 4 | 311 |
| Dnttip1 | Q99LB0 | 442 (37) | 8 (1) | 167 | 3 | 177 | 3 | 262 |
| Sec13 | Q9D1M0 | 498 (58) | 7 (1) | 207 | 5 | 76 | 2 | 260 |
| Kpna2 | P52293 | 292 (90) | 5 (1) | 260 | 5 | 226 | 3 | 259 |
| Dpf2 | Q61103 | 244 | 3 | 145 | 3 | 375 (148) | 8 (4) | 255 |

[a]Mascot score for the specified protein in the Nanog sample, purified by FLAG affinity. Mascot score for the specified protein in the corresponding control purification, if present, is in parentheses.
[b]Number of identified unique, nonredundant peptides for the specified protein in the Nanog sample. Number of identified unique peptides in the control purification is in parentheses.

region were generated (Figure 5A). Co-immunoprecipitation analyses show that while a Sox2 mutant truncated after residue 233 retained the ability to interact with Nanog, Sox2 mutants with deletion of residues between 205 and 233, or truncated after residue 212 were unable to interact with Nanog (Figure 5B). These analyses identify a critical Nanog-interacting region in Sox2 between residues 212 and 233. This sequence is highly enriched for hydroxyamino acids (12/21 residues) and, like the WR of Nanog, is devoid of acidic and basic side chains. Moreover, careful examination of this 21 amino-acid region highlighted three repeats of the sequence S X T/S Y that may be responsible for mediating the interaction with Nanog. To determine the potential importance of these motifs for the Nanog–Sox2 interaction, additional truncations were made after residues 218 and 226, which truncate Sox2 after repeat 1 or 2, respectively. This indicates that repeat 1 is sufficient for interaction with Nanog but that together repeats 1 and 2 interact with Nanog with an efficiency approaching that of wild-type Sox2 (Figure 5B). To examine the sequences required on Sox2 in more detail, a series of point mutations were generated

within the repeats (Figure 6). Individual or combinatorial contributions of each of the three repeats to Nanog binding were initially examined (Figure 6A and B). Mutation of individual repeats suggests an order of importance for Nanog interaction of repeat 1 > repeat 3 > repeat 2 (Figure 6A). This is supported by analyses of the combinatorial mutants where mutation of repeats 1 + 3 almost entirely eliminates the ability of Sox2 to interact with Nanog (Figure 6B). Mutations of the amino acids at positions 1, 3 or 4 in all the three repeats were next analysed. Combined mutations at positions 1 and 3 had negligible effects (Figure 6C), implying that residues at these positions are not required for Nanog interaction. In contrast, the combined mutation of the tyrosines at position 4 indicates that these residues play a key role in the interaction with Nanog (Figure 6C). Together, these experiments suggest that the tyrosines are the residues directly interacting with the Nanog WR, with the tyrosines in repeats 1 and 3 being more important in this regard than the tyrosine in repeat 2.

The Sox2 tyrosine residues could interact with Nanog partially via the hydroxyl, the phenyl ring or both. Since

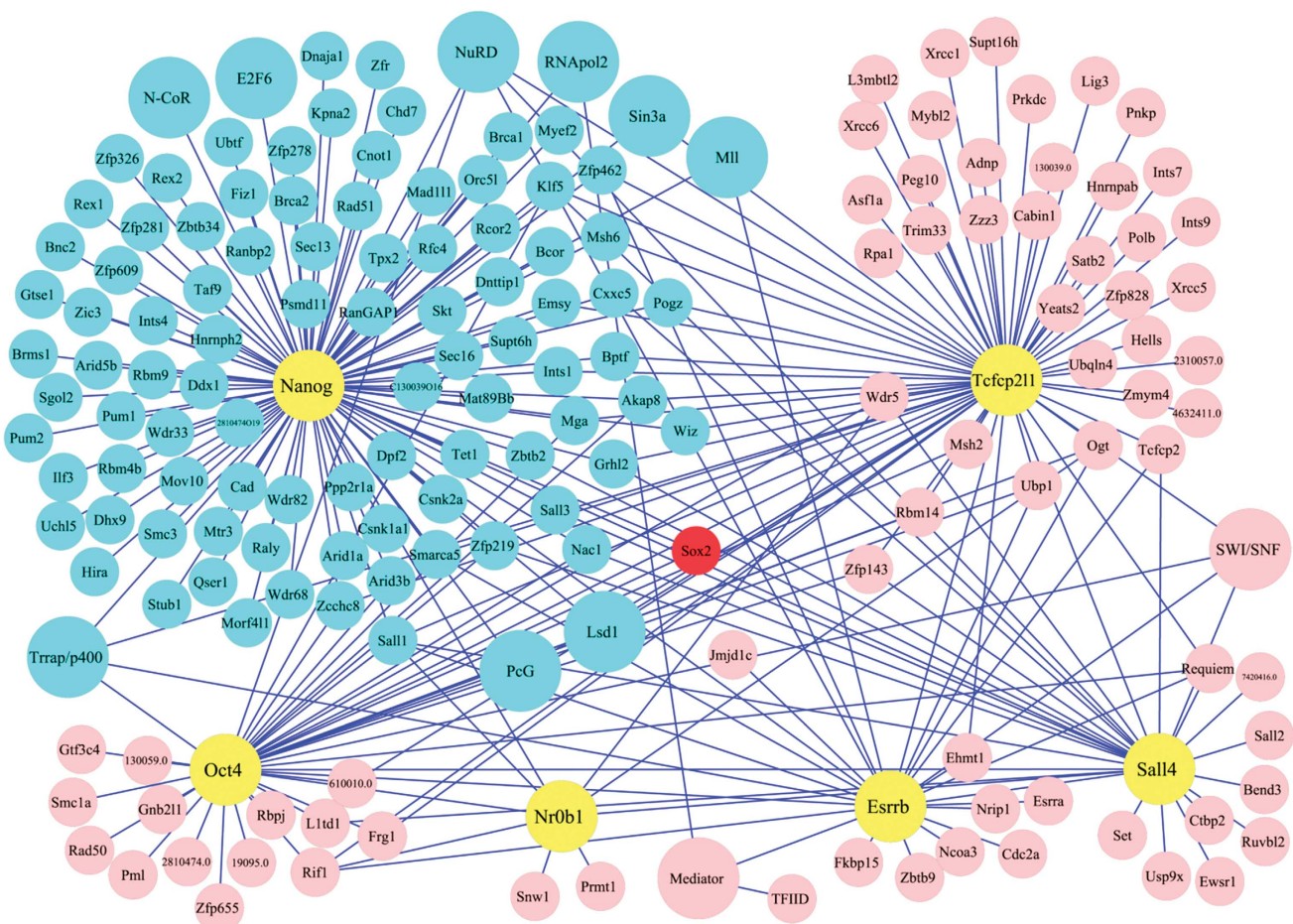

**Figure 2** The Nanog interaction network. The Nanog interactome (pale blue) as part of a larger interactome encompassing the interactions of Oct4, Esrrb, Sall4, Nr0b1 and Tcfcp2l1 (van den Berg *et al*, 2010).

the tryptophans in WR are critical for the Nanog/Sox2 interaction this raises the hypothesis that hydrophobic stacking of the aromatic rings in the Sox2 tyrosines and the Nanog WR tryptophans mediate the interaction. If these were the case, then the tyrosines hydroxyl groups should be unimportant for the interaction between Nanog and Sox2. To test this hypothesis, the tyrosines were mutated to phenylalanine. The direct comparison of the interaction between $(HA)_3$Nanog and $(Flag)_3$Sox2:YYY>A or $(Flag)_3$Sox2:YYY>F by co-immunoprecipitations clearly shows that substitution of the tyrosine residues with phenylalanines rescues the Nanog interaction, indicating that it is the benzene ring of these amino-acid residues that is required for the interaction to occur (Figure 6D).

### Identification of Nanog/Sox2 binding motif *in vitro*

To investigate possible DNA sequences bound by the Nanog/Sox2 complex, $(His)_6$-tagged Nanog and unmodified Sox2 were co-expressed in *E. coli* for use in Systematic Evolution of Ligands by Exponential Enrichment (SELEX). As controls, MBP-Nanog and $(His)_6$-Sox2 were expressed individually. Purification from bacterial lysate containing co-expressed proteins on a nickel column followed by elution with imidazole yielded two proteins of the expected size for Nanog and Sox2. These were recognized by α-Nanog and α-Sox2 antibodies (Figure 7A), with N-terminal sequencing establishing the identities of the two bands as Nanog and Sox2. The

Nanog–Sox2 interaction is robust, since the proteins co-purify through subsequent ion exchange (Figure 7B). The Nanog–Sox2 complex bound to the Ni-agarose, MBP-Nanog bound to amylose resin and $(His)_6$-Sox2 bound to Ni-agarose were used for SELEX, the bound oligonucleotides cloned and the sequences determined (Figure 7C) used to derive the motifs shown (Figure 7D). The motif obtained from Nanog alone has a TAAT core sequence followed by CG, consistent with the motif obtained previously by SELEX (Mitsui *et al*, 2003) and the nucleotide preferences of the isolated Nanog HD in EMSAs (Jauch *et al*, 2008). Sox2 also gives a motif highly similar to that determined by SELEX (CA/TTTGA/T) (Harley *et al*, 1994; Maruyama *et al*, 2005). The motif obtained from the Nanog/Sox2 complex is bipartite with bases 10–15 similar to the motif obtained by us and others for Sox2 alone (Harley *et al*, 1994; Maruyama *et al*, 2005) and bases 5–7 showing similarity to the central core of the Nanog motif identified by SELEX (TAAT) in this work and by others (Mitsui *et al*, 2003). However, the published Nanog motif has a high degree of confidence over a four base sequence (TAAT) while the Nanog–Sox2 binding sequence shows high certainty for only three bases (TAA) with the preference for the 3′-flanking CG no longer apparent. This difference may reflect an alteration in the binding specificity of Nanog when in complex with Sox2. Interestingly, the SELEX motif shows high similarity to a Nanog/Sox2 motif identified by *de novo* methods from

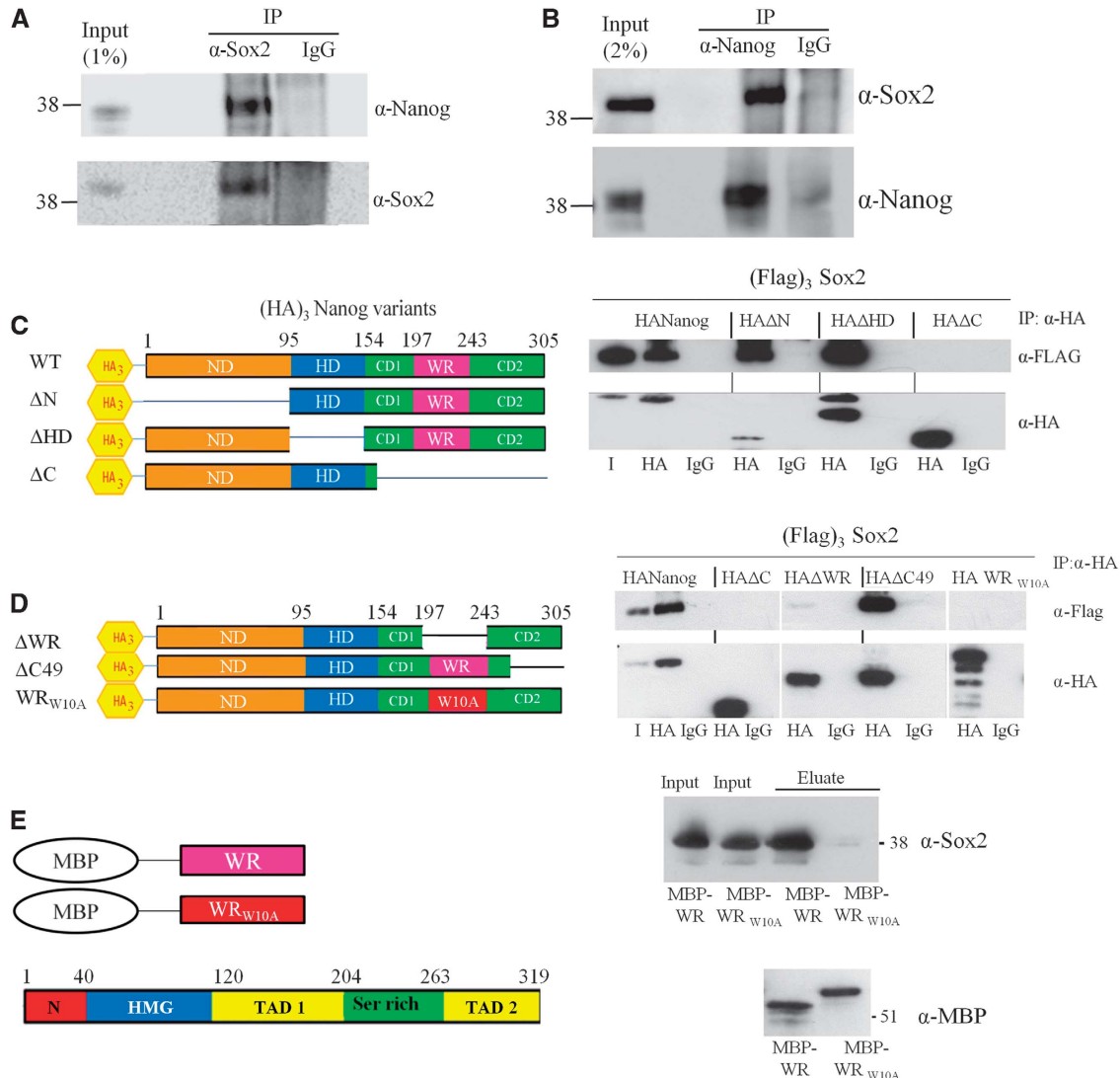

**Figure 3** Mutational analysis of the Sox2-interaction domain in Nanog. (**A**) Co-immunoprecipitation of endogenous Sox2 and Nanog from E14Tg2a nuclear extract. Immunoprecipitation was performed with Sox2 antibody and immunoblot probed with anti-Nanog or anti-Sox2 antibodies. (**B**) Co-immunoprecipitation of endogenous Nanog and Sox2 from E14Tg2a nuclear extract. Immunoprecipitation was performed with Nanog antibody and immunoblot probed with anti-Sox2 or anti-Nanog antibodies. (**C**) Left, schematic representation of the (HA)₃Nanog constructs. Right, co-immunoprecipitations of Nanog variants with Sox2. (FLAG)₃Sox2 and (HA)₃Nanog deletion mutants were transfected into E14/T cells. Immunoprecipitations were performed with an HA antibody as indicated and immunoblots probed with anti-FLAG or anti-HA antibodies. I is 1% of input. (**D**) Left, schematic representation of the (HA)₃Nanog constructs. Right, co-immunoprecipitations of Nanog variants with Sox2. (FLAG)₃Sox2 and (HA)₃Nanog deletion mutants were transfected into E14/T cells. Immunoprecipitations were performed with an HA antibody as indicated and immunoblots probed with anti-FLAG or anti-HA antibodies. I is 1% of input. (**E**) Left, Sox2, co-expressed in *E. coli* with either a Maltose Binding Protein-tryptophan repeat (MBP-WR) fusion protein or a Maltose Binding Protein-tryptophan repeat in which all the tryptophans were mutated to alanine (MBP-WR_{W10-A}). Right, MBP fusion proteins and associated proteins were purified on amylose resin, subjected to SDS–PAGE and immunoblots probed with Sox2 or MBP antibodies. Source data for this figure is available on the online supplementary information page.

ChIP-Seq data (Hutchins *et al*, 2013), which notably also contains a 2-bp gap between the major binding nucleotide groups (Figure 7D). Therefore, a combined motif was generated and used to search available ChIP-Seq data sets. Analysis of three independent ChIP-Seq data sets (Chen *et al*, 2008b; Marson *et al*, 2008; Whyte *et al*, 2013) identified 3257 Nanog/Sox2 overlapping peaks, which are common to the three data sets (out of a total of 16 454 from all Nanog/Sox2 overlapping peaks in the three data sets). Of these 3257 high confidence peaks, 29.1% (948 peaks) contain the motif. The motif occurs in a significantly smaller fraction of the Nanog only or Sox2 only peaks (4898 peaks out of a total of 31 271 peaks (15.7%; hypergeometric *P*-value $< 1 \times 10^{-10}$). Examples of

occurrences of the motif relative to the nearest gene are shown (Figure 7E; Supplementary Table II).

**The Nanog–Sox2 interaction is critical for Sox2 function**
To investigate the functional significance of the interaction between Nanog and Sox2, we took advantage of ES cells carrying a conditional *Sox2* knock-out allele (*Sox2*CKO). In this cell line, one of the *Sox2* alleles is flanked by *loxP* sites (Favaro *et al*, 2009), while the other *Sox2* allele has been replaced with a β-geo cassette (Zappone *et al*, 2000; Avilion *et al*, 2003). These cells also have a constitutively expressed CreER^T2^-IRES-Puro transgene integrated randomly in the genome. Upon addition of tamoxifen, CreER^T2^ is

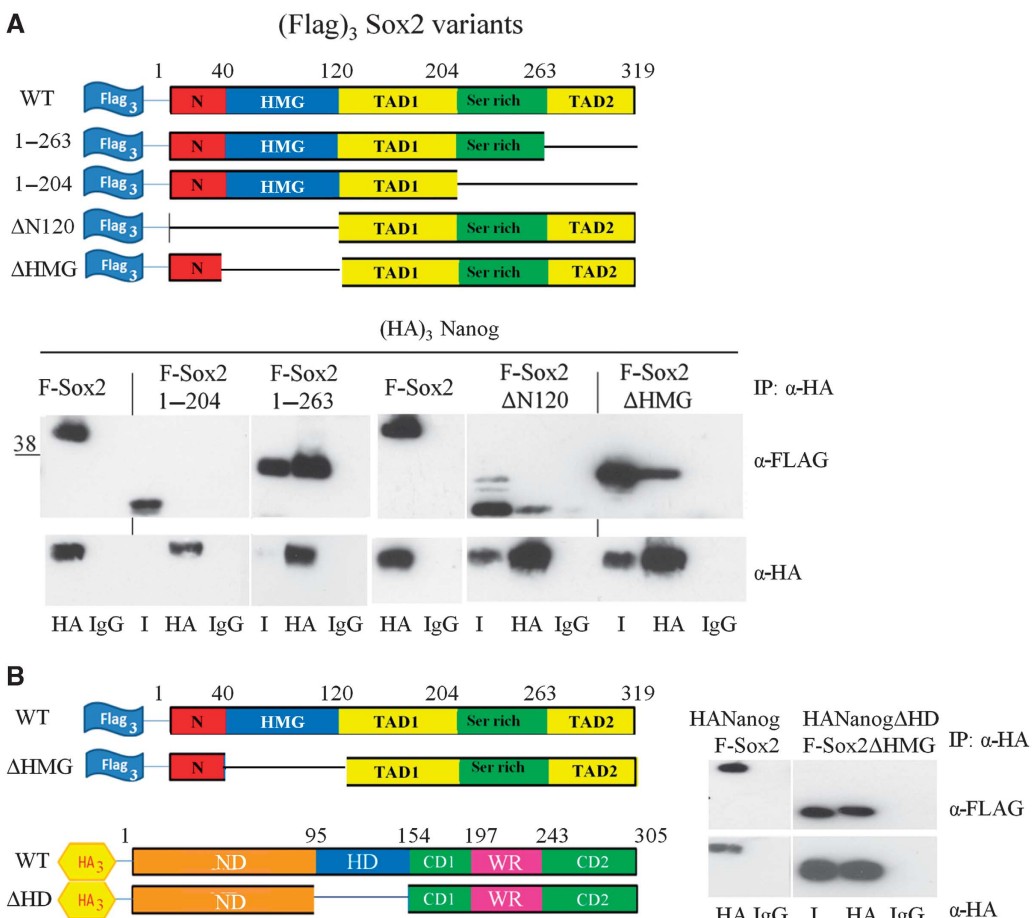

**Figure 4** The serine-rich domain of Sox2 interacts with Nanog. (**A**) Top, schematic representation of the (FLAG)₃Sox2 constructs. Bottom, (HA)₃Nanog and the indicated (FLAG)₃Sox2 deletion mutants were transfected into E14/T cells and immunoprecipitations were performed with an HA antibody as indicated and immunoblots probed with anti-FLAG or anti-HA antibodies. I is 1% of input. (**B**) Left, schematic representation of the (FLAG)₃Sox2 and (HA)₃Nanog constructs used to assess the DNA independence of the Nanog–Sox2 interaction. Right, E14/T cells were transfected with the indicated vectors. Immunoprecipitations were performed with an HA antibody as indicated and immunoblots probed with anti-FLAG or anti-HA antibodies. I is 1% of input. Source data for this figure is available on the online supplementary information page.

translocated to the nucleus and excises the *Sox2* gene between the *loxP* sites (Figure 8A). As ES cells from which Sox2 activity has been removed are unable to self-renew and differentiate into trophectoderm-like cells (Masui *et al*, 2007), this cell line was used to test whether Sox2 mutant molecules impaired in Nanog binding could rescue the Sox2 null phenotype (Figure 8B). *Sox2*CKO cells expressing a GFP control plasmid completely differentiate upon Tamoxifen treatment (Figure 8B). As expected, cells transfected with an unmutated (Flag)₃Sox2 cDNA rescued this differentiation phenotype. In contrast, cells expressing (Flag)₃Sox2:YYY>A transgene showed a decrease in self-renewal activity with 50% fewer undifferentiated colonies compared to wild-type Sox2 (Figure 8C). In accordance with the interaction data, expression of (Flag)₃Sox2:YYY>F fully rescued the differentiation phenotype (Figure 8C). To examine the possibility that the reduced colony formation by the Sox2:YYY>A cells was due to a reduced expression level, an immunoblot for Sox2 was performed. However, the amount of Sox2 expressed is comparable between Sox2:YYY>A and other lines and does not differ from the endogenous Sox2 level expressed by the parental line (Figure 8D). These data suggest that the interaction with Nanog is a key component in the function of Sox2 in ES cell self-renewal.

To further investigate the effect of disrupting the Nanog–Sox2 interaction, the expression of genes present in the ChIP-Seq data sets was examined in cell lines expressing wild-type or mutant Sox2 (YYY>A). Of 13 genes analysed, 5 showed consistent differences by qRT-PCR when the Nanog/Sox2 complex was disrupted (Figure 8E). The genes that show altered expression include transcription factors reported to be important for ES cell identity (Rex1 and Klf5 (Shi *et al*, 2006; Parisi *et al*, 2010), the gene encoding the chromatin re-modelling protein Myst4 (Ura *et al*, 2011) as well as the cell-surface markers Ncam and Itga9 (Rugg-Gunn *et al*, 2012). In addition, Oct4, which does not contain the Nanog/Sox2 motif, did not change expression level in absence of a Nanog/Sox2 functional complex. It is therefore likely that the effect of disrupting the Nanog/Sox2 complex on self-renewal is a consequence of the misregulation of the genes controlled by the two proteins in complex.

## Discussion

By taking advantage of improved methodology (van den Berg *et al*, 2010) the Nanog interactome has been expanded to over 130 proteins which can be subdivided into a number of different categories (Table I; Supplementary Table I). Many

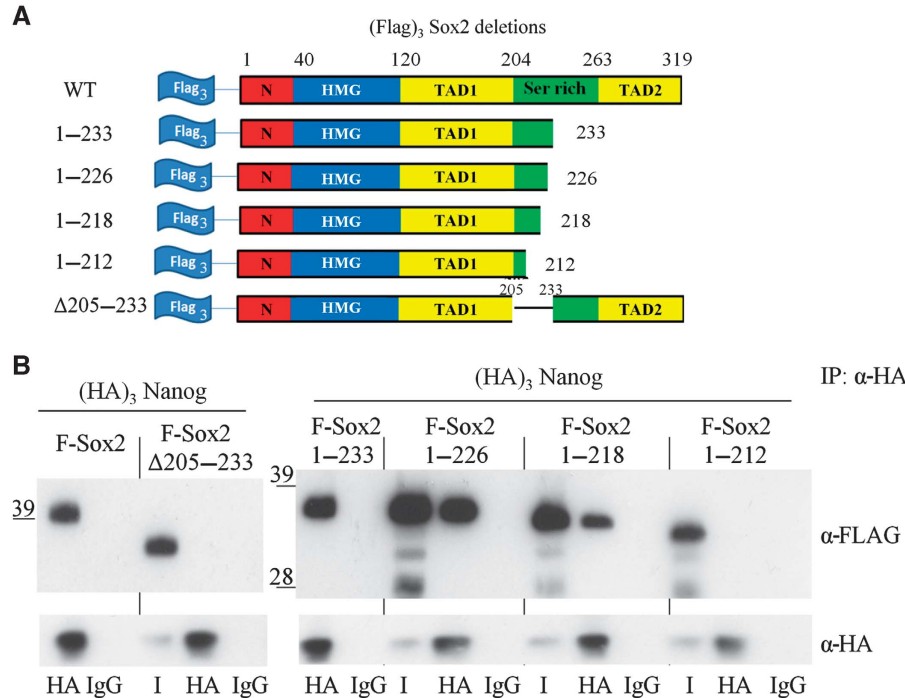

**Figure 5** A subregion of the Sox2 serine-rich domain interacts with Nanog. (**A**) Schematic representation of the (FLAG)₃Sox2 constructs used for mapping the minimal Sox2 region involved in the interaction with Nanog. (**B**) E14/T cells were transfected with (HA)₃Nanog and the indicated (FLAG)₃Sox2 mutants. Immunoprecipitations were performed with an HA antibody as indicated and immunoblots probed with an anti-FLAG or anti-HA antibodies. I is 1% of input. Source data for this figure is available on the online supplementary information page.

of the proteins identified in the interactome are components of large multi subunit complexes involved in chromatin modification, for several of which, all the known subunits are detected. Most of these are considered to be transcriptional repressors (NuRD, Polycomb Group protein (PcG), the atypical Polycomb complex E2F6, Sin3a and N-CoR) that bind to genomic sites adjacent to differentiation-specific genes to mediate repression (Jepsen and Rosenfeld, 2002; McDonel *et al*, 2009; Surface *et al*, 2010; Qin *et al*, 2012). Emerging evidence suggests that NuRD and PcG complexes are also found at sites that are actively transcribed (Brookes *et al*, 2012; Reynolds *et al*, 2012). How the NuRD complex is directed to target genes is not fully understood but Nanog and/or other NuRD-interacting transcription factors may target the complex to the relevant sites in the genome. In this respect, it is interesting that inducing Nanog protein results in enhanced binding of both Nanog and NuRD to the *Nanog* enhancer (Fidalgo *et al*, 2012).

Another proposed role for the chromatin modification complexes is to maintain repressed genes in a state that allows a rapid response to external cues. Evidence for this comes from the co-localization of enzymatically active PRC complexes and the paused form of RNA PolII at a large number of developmentally important genes (Brookes *et al*, 2012). This could allow alterations in the signalling environment to promptly increase the level of gene expression. The interaction of Nanog with both PRC and RNA PolII may reflect this poised state of some genes. The association with the chromatin modification machinery is common to transcription factors involved in maintenance of ES cell pluripotency (Wang *et al*, 2006; Liang *et al*, 2008; Pardo *et al*, 2010; van den Berg *et al*, 2010; Ding *et al*, 2012). However, the range of complexes binding to individual factors differs with SWI/SNF not directly connecting to

Nanog (this study; Wang *et al*, 2006) but interacting with other transcription factors (van den Berg *et al*, 2010). Recent data showing that Esrrb can substitute for Nanog function in ES cells (Festuccia *et al*, 2012) could in part be explained by the fact that Esrrb and Nanog bind to a number of the same chromatin modification complexes.

The Nanog interactome includes a number of proteins that have not previously been identified in an ES cell transcription factor interactome. In addition to TET-1, which has also been shown to interact with Nanog (Costa *et al*, 2013), these include the RNA processing proteins Ilf3, Rbm9, Pum1/2 and the transcription factors Zfp326, Arid5b, Zfp609. Examining the function of these molecules and the significance of their interaction with Nanog will provide further detail on how the extensive protein interaction network functions to control pluripotency.

In this study, we have focussed on the interaction between Nanog and Sox2 because of the central role of these proteins in the pluripotency gene regulatory network. Sox2 has been shown to interact with another key pluripotency factor, Oct4 by interaction of side chains within the DNA binding domains (Ambrosetti *et al*, 1997, 2000). In the case of Nanog and Sox2, interaction occurs through sequences outwith the DNA binding domains. Nevertheless, the sequence of the SELEX motif suggests that this interaction results in a specific spatial relationship of DNA binding domains of both proteins on DNA. The sequence of Sox2 that mediates interaction with Nanog is a triple repeat of the sequence S X S/T Y. Experiments analysing Sox2 mutants for their ability to rescue ES cells from differentiation induced by Sox2 deletion demonstrate the importance of the interaction of Nanog with Sox2. Mutation of the tyrosines in the S X S/T Y motifs to alanines reduces the formation of undifferentiated

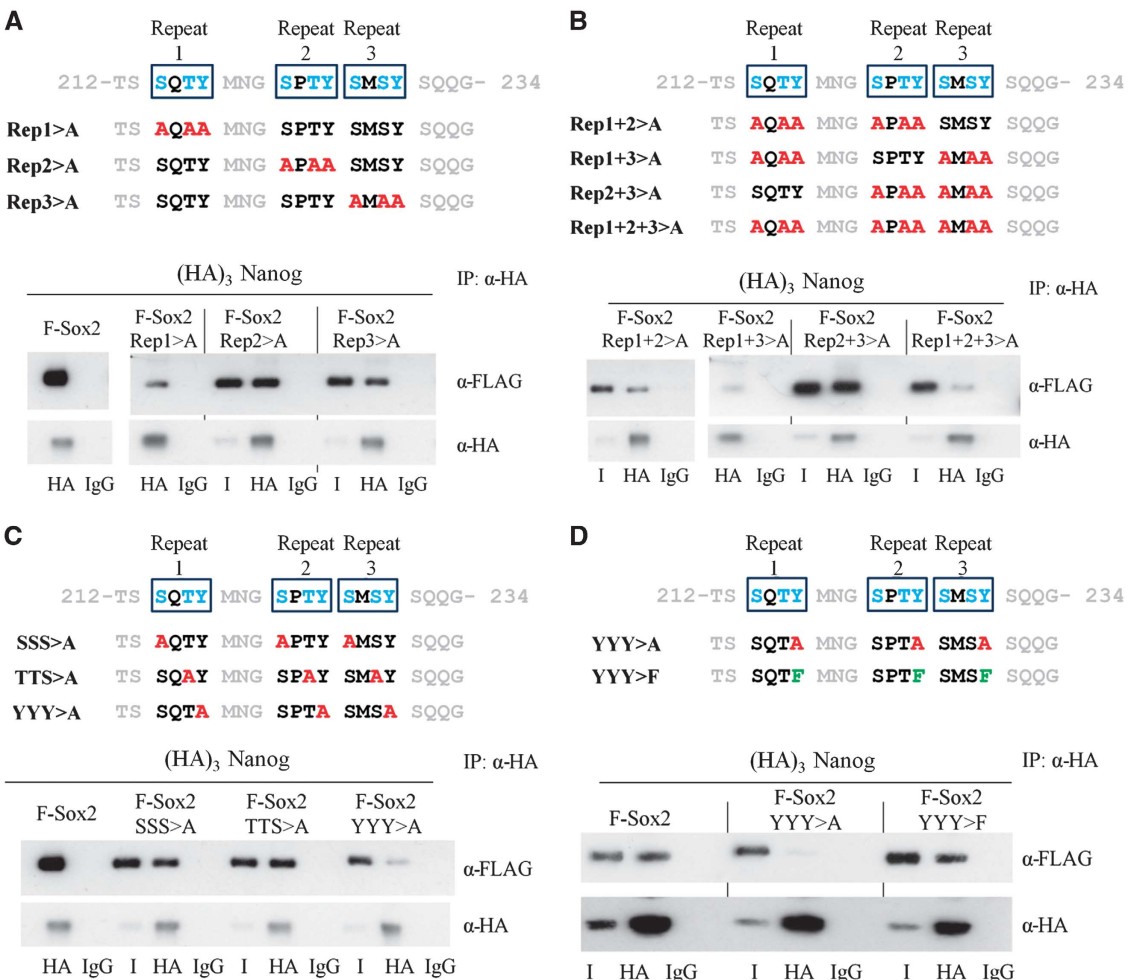

**Figure 6** Identification of amino-acid residues within Sox2$_{(213–233)}$ interacting with Nanog. (**A**) Top, schematic representation of hydroxyamino acid mutations in repeats 1, 2 or 3 in Sox2. Bottom, E14/T cells were transfected with (HA)$_3$Nanog and the indicated (FLAG)$_3$Sox2 mutants. Immunoblots of the HA immunoprecipitates were analysed by immunoblotting with an anti-FLAG or an anti-HA antibody. I is 1% of input. (**B**) Top, schematic representation of the combinatorial mutations of the hydroxyamino acids in repeats 1, 2 and 3 of Sox2. Bottom, E14/T cells were transfected with (HA)$_3$Nanog and the indicated (FLAG)$_3$Sox2 mutants. Immunoblots of the HA immunoprecipitates were analysed by immunoblotting with an anti-FLAG or an anti-HA antibody. I is 1% of input. (**C**) Top, schematic representation of the mutations of the hydroxyamino acids in positions 1, 3 or 4 of repeats 1, 2 and 3 of Sox2. Bottom, E14/T cells were transfected with (HA)$_3$Nanog and the indicated (FLAG)$_3$Sox2 mutants. Immunoblots of the HA immunoprecipitates were analysed by immunoblotting with an anti-FLAG or an anti-HA antibody. I is 1% of input. (**D**) Top, schematic representation of the mutations of the hydroxyamino acids in position 4 of repeats 1, 2 and 3 of Sox2. Bottom, E14/T cells were transfected with (HA)$_3$Nanog and the indicated (FLAG)$_3$Sox2 mutants. Immunoblots of the HA immunoprecipitates were analysed by immunoblotting with an anti-FLAG or an anti-HA antibody. I is 1% of input. Source data for this figure is available on the online supplementary information page.

ES cell colonies to 50% of the level achieved using a non-mutant Sox2 cDNA in the absence of any difference in protein levels expressed by the transgene. Therefore, the 50% drop in undifferentiated colonies observed in the presence of Sox2:YYY>A is a result of the misregulation of Nanog/Sox2 gene targets. The use of the SELEX motif identified as a Nanog/Sox2 target sequence together with a previously published *de novo* target sequence (Hutchins *et al*, 2013) allowed potential target genes of the Nanog/Sox2 complex to be identified. A number of these genes show altered expression upon abrogation of the Nanog/Sox2 interaction (e.g., Ncam, Itga9, Klf5 and Myst4). However, not all the genes tested are sensitive to loss of the interaction between Nanog and Sox2 (Supplementary Table II). This could suggest that in such cases the hydrophobic interaction of Nanog and Sox2 proteins is not required for chromatin binding, or that

only in some cases is the associated gene sensitive to disruption of the interaction. The latter is reminiscent of our finding that only a subset of loci that bind Nanog respond to the presence of Nanog by modulating expression of a nearby gene (Festuccia *et al*, 2012).

In ES cells, composite Oct/Sox binding sites have been proposed to be redundantly regulated by Sox4, Sox11 and Sox15 (Masui *et al*, 2007). However, this redundancy does not extend to blockade of differentiation caused by Sox2 deletion. Consistent with this, Sox4, Sox11 and Sox15 are not present in the Nanog interactome and none of these Sox proteins contains a sequence that matches the S X S/T Y motif.

The three copies of the S X S/T Y motif in Sox2 occur within a 15-residue sequence in which 9 residues are hydroxyamino acids. Despite this preponderance of hydroxyamino acids, it is the aromatic rings of the tyrosine residues that are

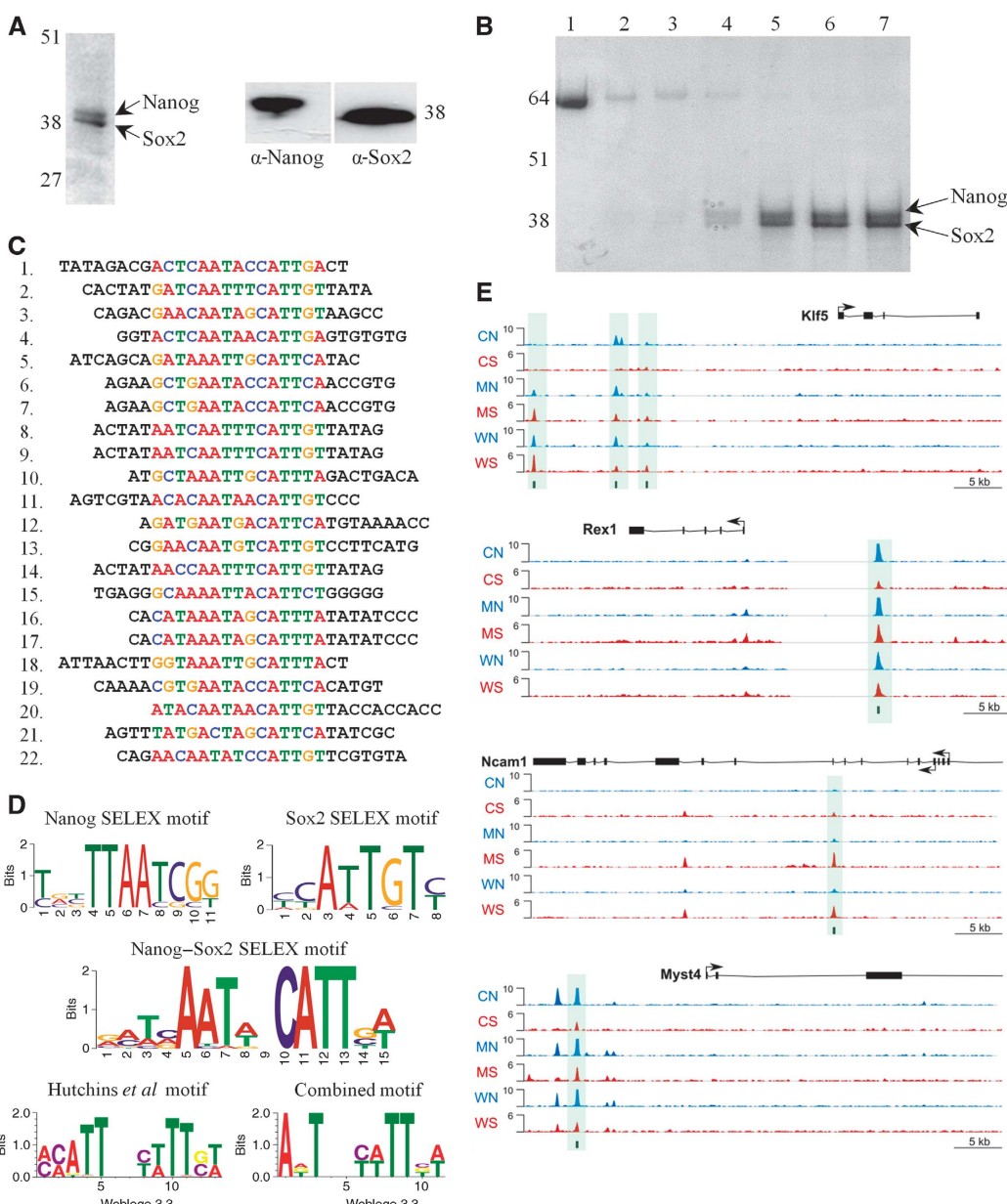

**Figure 7** Identification of a Nanog/Sox2 motif by SELEX. (**A**) Left, a Coomassie-stained SDS–PAGE gel of the imidazole eluate from the Ni-agarose purification of lysate from *E. coli* expressing His$_6$-Nanog and Sox2, showing the major two bands identified as containing Nanog (upper) and Sox2 (lower) by immunoblot analysis, as indicated on the right-hand panel. (**B**) Coomassie-stained SDS–PAGE gel of fractions from ion-exchange purification of Nanog/Sox2. Lanes 1–3 are protein from flow-through fractions and lanes 4–7 are from the eluted peak. (**C**) Sequence of 22 oligonucleotides that contribute to the motif generated by the *de novo* discovery program MEME. (**D**) Top panel, SELEX motifs generated for Nanog and Sox2 expressed individually from a total of 19 (Nanog) and 15 (Sox2) sequences submitted to MEME; middle panel, SELEX motif generated for Nanog/Sox2 complex from 38 sequences submitted to MEME; bottom panel, representation of the *de novo* Nanog/Sox2 motif (Hutchins *et al*, 2013) and the combined motif from SELEX sequence for Nanog/Sox2 and *de novo* Nanog/Sox2 motif. Motifs in the bottom panels were generated with Web Logo 3.3. (**E**) Nanog and Sox2 ChIP-seq peaks located near the transcriptional start sites of Zfp42, Klf5, Ncam1 and Myst4. The peaks that contain the Nanog/Sox2 motif are highlighted in the shaded box; Nanog (N) and Sox2 (S) peaks in data sets from Chen (C), Marson (M) and Whyte (W) data sets. Source data for this figure is available on the online supplementary information page.

critical mediators of the interaction with Nanog. This conclusion is derived from the fact that alanine substitution of all three serines at position 1 of the repeats or all three serines/threonines at position 3 of the repeats allowed continued efficient binding to Nanog, whereas alanine substitution of all three tyrosines decreased the Nanog interaction severely. Moreover, the fact that the Nanog interaction could be rescued when the tyrosines were substituted by

phenylalanines indicates that the tyrosine hydroxyl groups are not required for the interaction and is highly suggestive that the two proteins interact by stacking of the aromatic rings. This is consistent with the fact that tyrosine and tryptophan residues cluster at protein–protein interaction 'hot spots' (Bogan and Thorn, 1998; DeLano, 2002). Functionally relevant stacking of tryptophan and tyrosine residues has also been demonstrated in the

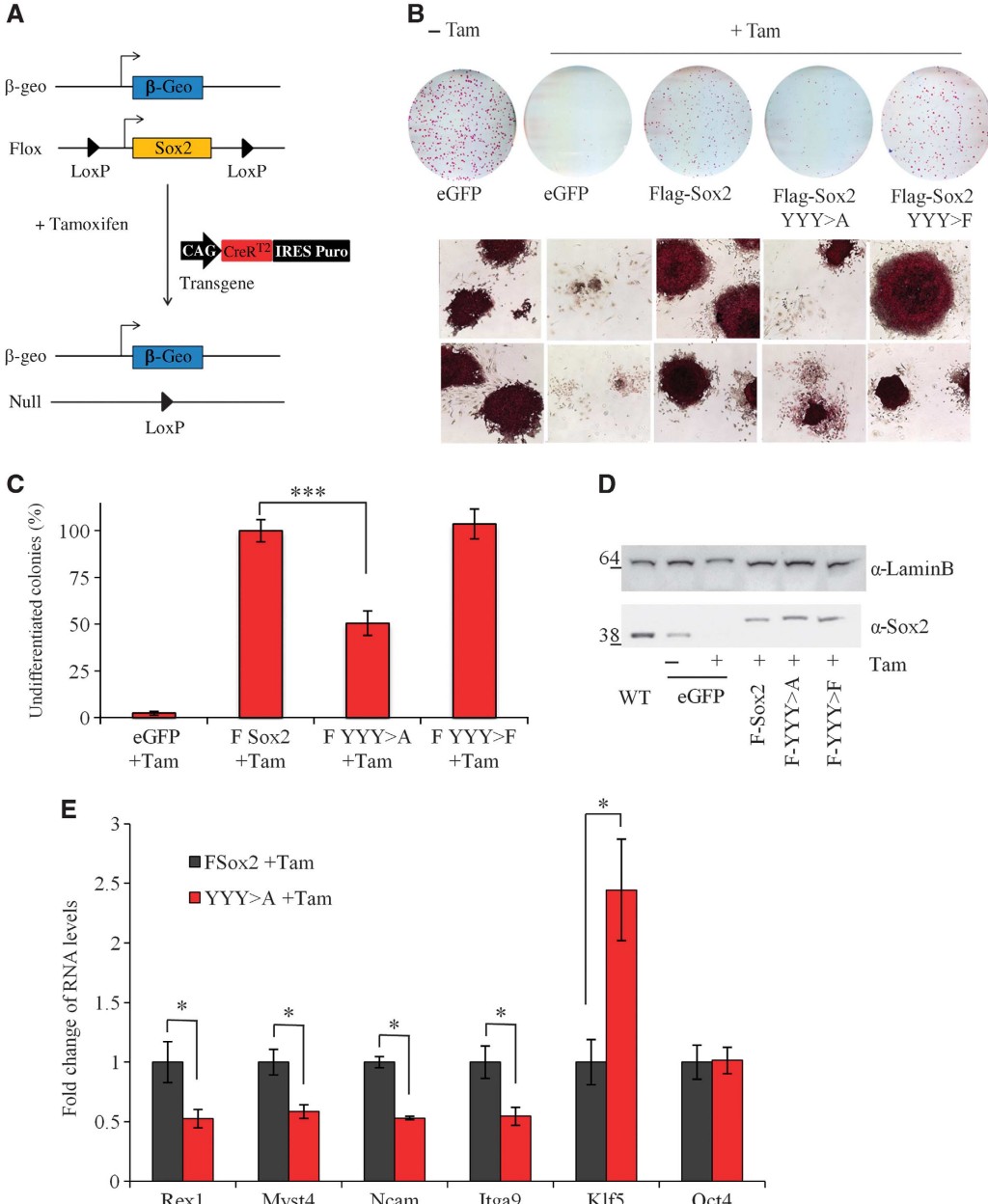

**Figure 8** The aromatic tyrosine side chains of the Nanog-interaction domain are required to fully rescue differentiation of Sox2 null ES cells. (**A**) Scheme of the Sox2 conditional knock-out (Sox2CKO) cell line. (**B**) Top, alkaline phosphatase staining of the Sox2 null rescued cells. Sox2CKO was transfected with the indicated vectors, selected in hygromycin and cultured in the presence or absence of tamoxifen. Cells were grown at clonal density for 7 days before staining. Bottom, two representative fields of colonies obtained for the indicated treatments. (**C**) Quantification of the data from **B**. The number of alkaline phosphatase-positive colonies formed following transfection with the indicated vector was calculated as a proportion of the number obtained by transfection with (Flag)₃Sox2, which was set as 100%. Error bars, s.e.m. of at least four independent experiments. ***Differences between FSox2 and FSox2:YYY>A by Mann–Whitney test ($P<0.001$). (**D**) Expression level of Flag-Sox2 variants in clonal assay. Comparison of Sox2 expression in E14Tg2a (WT) with *Sox2* CKO populations expressing eGFP, Flag-Sox2, FlagSox2:YYY>A, FlagSox2:YYY>F, plated at clonal density for 7 days. (**E**) Expression levels of Nanog/Sox2 target genes. Expression levels of Rex1, Myst4, Ncam, Itga9, Klf5 and Oct4 in cells plated at clonal density, treated with tamoxifen for 12 h and grown for 7 days before RNA extraction. The expression level in Sox2CKO F-Sox2 + tamoxifen was set to 1. Error bars, s.e.m. of three biological replicates. *Differences between FSox2 and FSox2:YYY>A by two-tailed *t*-test ($P<0.05$). Source data for this figure is available on the online supplementary information page.

formation of an aromatic gate in apo flavodoxin (Genzor *et al*, 1996), in the regulation of galactose oxidase activity (Rogers *et al*, 2007) and, of particular relevance to this study, in the interlocking of tyrosine and tryptophan residues at the interaction interface of human nuclear receptor pregnane X receptor (PXR) that mediates protein homodimerization (Noble *et al*, 2006).

The lack of a requirement for the hydroxyl groups on Sox2 for the Nanog interaction is underscored by experiments using bacterially expressed recombinant proteins that demonstrate that the Nanog–Sox2 interaction occurs in the absence of post-translational modifications. However, this does not mean that post-translational modifications might not affect the interaction between Nanog and Sox2. The interaction between the

two proteins occurs through polypeptide stretches devoid of strongly charged amino-acid side chains. Potential modification of hydroxyl groups on the Sox2 interaction surface, whether on the tyrosine or on the neighbouring serine and threonine residues, would introduce charged moieties that would be expected to interfere with the interaction between the hydrophobic interacting residues. Moreover, recent work indicates that hydroxyl groups on Sox2 can also be modified by addition of *N*-acetylglucosamine, although the effect on Sox2 function is unclear (Jang *et al*, 2012). The fact that Nanog interacts with proteins that mediate post-translational modifications such as phosphorylation and ubiquitination is consistent with the observation that Nanog is phosphorylated (Yates and Chambers, 2005; Moretto-Zita *et al*, 2010) and ubiquitinated (Moretto-Zita *et al*, 2010). In addition, Nanog partners could also be affected by such modifications because of physical proximity to the relevant enzymes. The role for these modifications and how they influence interactions between transcription factors and/or transcription factor function in ES cells is an important area for future investigation.

The high number of Nanog-interacting proteins identified in this study suggest that Nanog acts as a 'hub' protein (Han *et al*, 2004; Mullin and Chambers, 2012). The ability of individual partner proteins to interact with a hub protein like Nanog depends on the affinity of the interaction and the availability of the binding sites on both the hub protein and the partner, as has been discussed previously (Han *et al*, 2004; Mullin and Chambers, 2012). Since both competitive and non-competitive interactions are simultaneously possible, it will be important to determine which factors compete for the same regions of Nanog. Of particular relevance will be whether factors that bind through the WR interact through a precise subregion of the WR or if there is variability in the exact sequence bound by a specific partner. To date, only Sox2 and Nac1 have been demonstrated to interact directly with the WR. Loss of the tryptophans of the WR has also been demonstrated to abrogate the interaction of Nanog with Sall4, Nr0b1, Zfp198 and Zfp281 (Wang *et al*, 2008) but a direct interaction has not yet been shown for these proteins. In the situation where multiple factors bind the WR it is possible that binding of one factor could increase the affinity of another protein for interaction with an adjacent site in the same region resulting in co-operative binding of two or more factors. A clear potential example of this could be Nac-1, which has been reported to bind the C-terminal WR subunit (Ma *et al*, 2009). It is possible that both competitive and non-competitive binding to distinct sites on Nanog occurs simultaneously, allowing the assembly of large, functionally active complexes. An additional level of complexity arises from the possibility that the Nanog/Sox2 interaction may occur with either monomeric or dimeric Nanog (Mullin *et al*, 2008; Wang *et al*, 2008). Potential mechanisms that affect the Nanog dimerization equilibrium, such as covalent modifications, could thereby play an important part in regulating interactions and subsequent downstream events.

## Materials and methods

### ES cell culture
Mouse ESC lines were cultured on gelatin-coated dishes without feeders in GMEM/β-mercaptoethanol/10% FCS/LIF (GMEMβ/FCS/ LIF) as described (Smith, 1991). Nanog null RCNβH-B(t) cells have been described (Chambers *et al*, 2007): briefly, these cells have an IRES-HygromycinR-pA or an IRES-βgeo-pA replacement of *Nanog* sequences from intron I through to the 3'UTR. Sox2 conditional knock-out cells were obtained by re-targeting ES cells heterozygous for a *Sox2*$^{flox}$ allele (Favaro *et al*, 2009) with a *Sox2*-β-geo 'knock-in' targeting vector (Zappone *et al*, 2000; Avilion *et al*, 2003). This was followed by stable transfection of a pPyCAG-CreER$^{T2}$IP construct (Figure 8A). Puromycin-resistant clones were screened for efficient deletion of the *Sox2*$^{flox}$ allele following tamoxifen treatment to select the Sox2CKO clone used here.

E14Tg2a (Flag)$_3$-Nanog and RCNβH-B(t) (Flag)$_3$-Nanog cells were generated by electroporating E14Tg2a (Hooper *et al*, 1987) and RCNβH-B(t) cells with pPyCAG (Flag)$_3$NanogIP (Mullin *et al*, 2008) linearized with *Sca*I. Electroporated cells were plated in GMEMβ/ FCS/LIF and after 30 h, 1 μg/ml puromycin (Sigma, P9620) was added. Medium was replaced every 2 days and after 12 days, puromycin-resistant colonies were picked and Nanog expression levels were determined by immunoblotting with α-Nanog antibody (Chambers *et al*, 2007). Sox2CKO cells expressing (Flag)$_3$-Sox2, (Flag)$_3$-Sox2:YYY>A, (Flag)$_3$-Sox2: YYY>F or eGFP control were generated by electroporating Sox2CKO cells with pPyCAG(Flag)$_3$-Sox2IH, pPyCAG(Flag)$_3$-Sox2:YYY>AIH or pPyCAG(Flag)$_3$-Sox2: YYY>FIH linearized with *Fsp*I. Electroporated cells were plated in GMEMβ/FCS/LIF medium and after 30 h, 100 μg/ml hygromycin B (Roche, 10843555001) was added. Medium was replaced every 2 days and after 12 days, hygromycin-resistant colonies were pooled to generate populations for rescue assays.

Colony-forming assays were as described (Chambers *et al*, 2003).

### Plasmids
(HA)$_3$-Nanog, (HA)$_3$-NanogΔN, (HA)$_3$-NanogΔHD, (HA)$_3$-NanogΔC, (HA)$_3$-NanogΔC49 and (HA)$_3$-NanogΔWR have been described (Mullin *et al*, 2008). (HA)$_3$-NanogWR$_{W10A}$ was generated by inserting synthetic DNA in which all 10 tryptophan codons were replaced by alanine codons between the two *Sex*AI sites in the Nanog ORF. Flag-Sox2 was generated by cloning a Sox2 PCR product between the *Bam*HI and *Not*I sites of pPyCAG(FLAG)$_3$IP. (Flag)$_3$-Sox2 1-263, (Flag)$_3$-Sox2 1-204, (Flag)$_3$-Sox2 1-233, (Flag)$_3$-Sox2 1-226, (Flag)$_3$-Sox2 1-218 and (Flag)$_3$-Sox2 1-212 were generated by PCR using a forward primer containing a *Xho*I site 5′ to the (Flag)$_3$ tag and reverse primers containing a stop codon at the desired position followed by a *Not*I site. (Flag)$_3$-Sox2 ΔHMG and (Flag)$_3$-Sox2 Δ205–233 were generated by PCR by overlap extension (Ho *et al*, 1989). PCR products were cloned between the *Xho*I and *Not*I sites in pPyCAGIP. (Flag)$_3$-Sox2 ΔN120 was generated by PCR using a forward bipartite primer, containing a *Bam*HI site, which anneals to the linker between the (Flag)$_3$ tag and the Sox2 sequence and a sequence starting at codon 121 and a reverse primer containing a *Not*I site after the Sox2 stop codon. The PCR product was cloned between the *Bam*HI and the *Not*I sites in pPyCAGIP(Flag)$_3$-Sox2 vector. The primers used for the above PCRs are listed in Supplementary Table III.

(Flag)$_3$-Sox2:Rep1>A, (Flag)$_3$-Sox2:Rep2>A, (Flag)$_3$-Sox2: Rep3>A, (Flag)$_3$-Sox2:Rep1+2>A, (Flag)$_3$-Sox2:Rep1+3>A, (Flag)$_3$-Sox2:Rep2+3>A, (Flag)$_3$-Sox2:Rep1+2+3>A, (Flag)$_3$-Sox2:SSS>A, (Flag)$_3$-Sox2:TTS>A, (Flag)$_3$-Sox2:YYY>A and (Flag)$_3$-Sox2:YYY>F were generated by introducing synthetic DNA encoding the desired sequences (Genscript) between *Age*I and *Cla*I sites in the Sox2 ORF. These two sites were introduced into Sox2 by silent mutation of C(693)>G (*Age*I) and C(810)>A, C(813)>G (*Cla*I) (NM_011443.3). The coding sequence of all plasmids generated was checked by sequencing for absence of unwanted mutations.

### Rescue experiments
Sox2CKO cell lines, stably expressing (Flag)$_3$-Sox2, (Flag)$_3$-Sox2:YYY>A, (Flag)$_3$-Sox2:YYY>F or eGFP were preselected for 3–5 days with 200 μg/ml G418 (PAA, P31-011), 0.75 μg/ml puromycin and 100 μg/ml hygromycin B. Cells were plated at clonal density ($3 \times 10^3$ cells/63 cm$^2$ dish) in hygromycin B (100 μg/ml) for 24 h. 4-OH Tamoxifen (1 μM) (Sigma, H7904) was added (12 h) to delete the endogenous Sox2 allele. Cells were cultured in 100 μg/ml hygromycin B for 14 days before alkaline phosphatase staining (Sigma, 86R). The rescue efficiency was determined from the ratio of alkaline-positive ES cell colonies (plus Tamoxifen/minus Tamoxifen).

### Protein purification

Preparation of nuclear extracts and purification of Flag-tagged proteins were performed as described (van den Berg *et al*, 2010). Briefly, nuclear extract was prepared from cells (Dignam *et al*, 1983) and Flag-tagged protein purified using 60 μl Flag-agarose beads per 1.5 ml of nuclear extract, during which samples were treated with 150 U/ml DNase Benzonase (4°C, 3 h) to decrease spurious protein purification due to DNA bridging. Nanog and interacting proteins were then eluted using Flag peptide (0.2 mg/ml). For production of proteins in *E. coli* MBP-WR/Sox2 and MBP-WR$_{W10>A}$/Sox2 were cloned into pET Duet (Novagen) and expressed in BL21(DE3) cells. MBP-tagged proteins were lysed in 10 mM Tris pH 8.0, 100 mM NaCl, passed over amylose resin (NEB, E8021S), washed and proteins eluted with 10 mM maltose. Co-purifying protein was detected by immunoblotting. For SELEX, Nanog was cloned into pMalc2e (NEB) in frame with MBP and expressed in BL21 cells by addition of 1 mM IPTG. Cells were lysed in 10 mM Tris pH 8.0, 200 mM NaCl and purified on amylose resin. Sox2 was cloned into pET15b (Novagen) and expressed in BL21(DE3) by addition of 1 mM IPTG. Protein was purified by lysing cells in 25 mM Tris pH 8.0, 30 mM imidazole, 500 mM NaCl and passing lysate over nickel resin (His-select, Sigma, P6611). For co-expression, Nanog and Sox2 were cloned into pET Duet, to encode (His)$_6$-Nanog and unmodified Sox2 and expressed in BL21(DE3) cells induced with 1 mM IPTG. Cells were lysed in 25 mM Hepes pH 7.6, 1 M NaCl, 5 mM imidazole and lysate incubated in batch mode with nickel resin. Ion-exchange purification of Nanog/Sox2 was performed at pH 7.6 on a 1-ml CM Sepharose FF column (GE Healthcare, 17-5056-01). Bound protein was eluted using a gradient of 0–1 M NaCl over 20 column volumes.

### Co-immunoprecipitations

For validation of Nanog interactors from E14Tg2a F-Nanog, 5 μg of Flag antibody (Sigma, F3165) or mouse IgG control (Santa Cruz, sc-2015) was added to 200 μl of nuclear extract and incubated (4°C, 3 h). Immunocomplexes were purified by addition of Protein G, washed in 20 mM Hepes pH 7.6, 10% glycerol, 100 mM KCl, 1.5 mM MgCl$_2$, 0.2 mM EDTA pH 8, 0.2% NP-40, 0.5 mM DTT and eluted in Laemmli buffer (Laemmli, 1970). For probing extracts from RCNβH-B(t) and RCNβH-B(t) (Flag)$_3$-Nanog, FLAG agarose (Sigma, A2220) was used to purify proteins. Complexes were eluted in Laemmli buffer and analysed by immunoblotting.

For the characterization of the Nanog–Sox2 interaction, E14/T cells were transfected using Lipofectamine 2000 (Invitrogen, 11668-019) with (HA)$_3$-Nanog or Nanog mutants and (Flag)$_3$Sox2 or Sox2 mutants and nuclear extract was prepared. In all, 5 μg of HA antibody (MMS-101P, Covance) was added (4°C, 3 h) and immunocomplexes purified as above. Immunocomplexes were fractionated on NuPage-Novex 10% Bis-Tris gels (Invitrogen, NP0301BOX) and co-immunoprecipitating proteins detected by immunoblotting using the following antibodies: α-Sall4 (gift of Matthias Treir), α-Mta2 (8106, Abcam), α-Sox2 (sc-17320, Santa Cruz), α-Nanog (A300-397A, Bethyl Laboratories), α-Nac-1 (29047, Abcam), α-RNAPolII (PB-7C2, Euromedex) and α-Flag antibody (Sigma, F3165).

### Immunoblots

Cells were lysed in 20 mM Hepes pH 7.6, 20% glycerol, 250 mM KCl, 1.5 mM MgCl$_2$, 0.2 mM EDTA, 0.5 mM DTT, 0.5% NP-40 and 1 × protease inhibitor cocktail (Roche). Protein extract was treated with 150 U/ml DNase Benzonase (Novagen) (30 min, 4°C), 60 μg of lysate run on SDS–PAGE (Laemmli, 1970), transferred onto nitrocellulose (Whatman® Protran®) and blots probed with antibodies diluted in 5% non-fat dry milk/TBS/0.01% Tween-20. Membranes were developed with Super-Signal West Pico (Pierce) and exposed to Hyperfilm (Amersham). Primary antibodies were anti-Nanog (Chambers *et al*, 2007), 0.5 μg/ml, anti-Sox2 (sc-17320, Santa Cruz) 0.2 μg/ml and anti-LaminB (sc-6216, cs-6217 Santa Cruz) 0.2 μg/ml.

### Protein interaction network criteria

Criteria for inclusion in Table I and Supplementary Table I as a Nanog-interacting protein are present in two out of the three purifications from E14Tg2a:F-Nanog and RCNβH-B(t):F-Nanog cells, with a Mascot score of >50 and at least three-fold higher than in the corresponding control experiment. The Mascot score is a statistical measure of confidence of correct identification of a protein from its peptides (Perkins *et al*, 1999).

### SELEX

Recombinant proteins used in SELEX assays were purified as described above. Purified protein identities were established by N-terminal sequencing (performed at the LIGHT Laboratories, Faculty of Biological Sciences, Leeds University by Edman degradation). Resin was washed with purification buffer supplemented with imidazole to 50 mM and used directly in SELEX. The oligo library contained 25 bp of random sequence flanked by ACGTG GATCCACTGACGG and GCTAGCGCCTCGAGACTTG. The double stranded library was synthesized by annealing the single-stranded library to a reverse primer and incubating in a Klenow fragment reaction. The initial round of SELEX consisted of incubation of 10 pmol of protein (on bead) with 20 pmol library in 20 mM Hepes pH 7.6, 200 mM KCl, 10% glycerol for 1 h. After extensive washing, bound oligonucleotides were eluted with imidazole containing buffer. PCR amplification of bound oligonucleotide was performed and the enrichment cycle repeated with 300 ng of total DNA. Five enrichment rounds were performed after which PCR products were cloned into TopoTA (Invitrogen). Individual clones were picked, DNA prepared and sequenced. Obtained sequences were submitted to MEME for motif searching (Bailey *et al*, 2009).

### Combined ChIP-seq and motif analysis

To identify joint Nanog–Sox2 (NS) DNA-binding events, enriched binding events ('peaks') based on three ChIP-seq experiments (Marson *et al*, 2008; Chen *et al*, 2008b; Whyte *et al*, 2013) were combined using *GeneProf* (http://www.geneprof.org; Halbritter *et al*, 2012). After centring peaks on ± 50 bp surrounding the highest point in the alignment, those overlapping in at least 1 bp were taken forward as joint NS peaks. A regular expression (A[TGA]T..[TC][AT]TT[GCT][AT]) was used to find occurrences of the NS motif. Peaks were linked to the closest transcription start site (max. distance ≤ 30 kb). The full analysis workflow is accessible at *gpXP_001309* (http://www.geneprof.org/show?id = gpXP_001309).

### Quantitative PCR

RNA was extracted with TRIZOL (Invitrogen), DNase treated (Qiagen) and reverse transcribed with SuperScriptIII (Invitrogen). (Q)PCRs were performed in 384-well plates with a 480 LightCycler (Roche) using LightCycler 480 SYBR Green I Master (Roche). All primer sequences are listed in Supplementary Table III.

### Supplementary data

Supplementary data are available at *The EMBO Journal* Online (http://www.embojournal.org).

## Acknowledgements

We are grateful to P Chambon for CreER$^{T2}$ and to M Trier for Sall4 antibody. Research in IC's laboratory was funded by The Wellcome Trust and the IC and ST laboratories by The Medical Research Council of the UK (including a studentship to AG) and the EU Framework 7 project 'EuroSyStem'; RP's laboratory was supported by a VIDI grant (NWO) and the Netherlands Institute of Regenerative Medicine network; SKN's laboratory was supported by grants from ASTIL Regione Lombardia (SAL-19 Ref no 16874), Telethon (GGP12152), Cariplo (2010-0673) and AIRC (IG-5801).

*Author contributions*: AG, NM and DC prepared cell lines and performed the biochemical analysis. AG and ZYT performed mutagenesis and analysis of Sox2. AF, JW and NM performed SELEX. JD and KB performed MS analysis. RAP provided advice on protein purification and MS assistance. RF and SKN prepared the Sox2 mutant cell line. AK, FH and SRT performed bioinformatics analyses. IC conceived the project and with NM and AG analysed the data. IC, NM and AG wrote the paper and with SKN, JD and RAP edited the manuscript.

## Conflict of interest

The authors declare that they have no conflict of interest.

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
