## [Review Process File · The EMBO Journal]

Manuscript EMBO-2013-84620

A direct physical interaction between Nanog and Sox2 regulates embryonic stem cell self-renewal

Alessia Gagliardi, Nicholas P Mullin, Zi Ying Tan, Douglas Colby, Anastasia Koussa, Florian Halbritter, Jason T Weiss, Anastasia Felker, Karel Bezstarosti, Rebecca Favaro, Jeroen Demmers, Silvia K Nicolis, Simon R Tomlinson, Raymond A Poot and Ian Chambers

Corresponding author: Ian Chambers, Edinburgh, University of

Review timeline:

Submission date:	30 January 2013
Editorial Decision:	18 February 2013
Revision received:	09 June 2013
Additional Correspondence:	12 June 2013
Accepted:	27 June 2013

Transaction Report:

Editor: Thomas Schwarz-Romond

1st Editorial Decision

18 February 2013

Thank you very much for submitting your study on the Nanog-interactome and the dissection of direct Sox2 interaction for consideration to The EMBO Journal editorial office.

As you will see, three expert referees emphasize the merits of the study. At the same time, they offer constructive suggestions to improve the significance of the presented dataset.

Specifically, all referees demand stronger functional support. Ref#1 limits this to better characterization of the pluripotency/differentiation potential of the engineered Nanog cells. Rather more critical, refs#2 and #3 request further reaching support for the relevance of the Nanog/Sox2 interaction. This should go beyond quantification of the mutant analyses and cover potential synergistic DNA-binding to alter transcriptional outputs as well as potential impact of Nanog-dimerization on Sox2 binding.

Convinced that you are in a very strong position to address these demands in a rather timely manner, I am delighted to invite submission of a thoroughly revised study for final assessment by some of the original referees.

Please do not hesitate to get in touch in case of further questions (preferably via E-mail).

Please be reminded that The EMBO Journal considers only one round of revisions and the ultimate decision on publication dependent on the outline and strength of the revised manuscript.

I am very much looking forward to your amended study and remain with best regards.

REFEREE REPORTS

Referee #1:

The paper by Gagliardi et al, "A direct physical interaction between Nanog and Sox2..." provides a comprehensive, proteomic view of partner proteins that interact with Nanog in mouse embryonic stem cells. One of the interacting partners revealed is Sox2. Given that Sox2 and Nanog are both important regulators of pluripotency, the authors went on to biochemically detail the physical interaction between these two transcription factors. They also show that the pluripotency function of Sox2 is compromised by loss of Nanog association.

This work is original and insightful as it shows the importance of combinatorial interactions between key transcription factors to execute an important biological function. Overall the experiments were performed with care, the data presented clearly, and the manuscript well written. However, few points should be addressed:

1. The proteomic analyses were done with high rigor, as previously advanced by the group, to assess other interactomes in ES cells. This paper provides too little detail to the reader without going to referenced papers. At least a brief description of the method is warranted here, particularly noting the use of DNase that precludes indirect associations through co-binding to DNA.
2. The F-Nanog cell line used should be better characterized with respect to retaining pluripotency traits-marker expression or differentiation potential should be provided to confirm that the engineered cells are still pluripotent.
3. Table I, II: "ZIC3" should be "Zic3"
4. Figure 1D: the "C F-Nanog" lane labels as are a bit confusing due to spacing.
5. Figure 2: It was difficult to quickly locate Sox2 in the diagram. Perhaps, given the importance of this factor in the paper, it could be color or bold highlighted
6. Figure 3C: The Sox2 schematic, in multi-colors, should have the domains labeled, consistent with Figure 4.
7. Figure 7B legend: please indicate the time point at which the colonies were scored.
8. The discussion section could be sharpened up a bit; currently a bit loose and wordy.

Referee #2:

In this well-written manuscript, Gagliardi et al. extend the protein interactome of a key self-renewal factor in mouse ES cells, Nanog. They demonstrate for the first time direct interaction with another core transcription factor, Sox2, and, in a series of co-immunoprecipitation experiments, reveal the motifs on both Nanog and Sox2 mediating this interaction. This is clearly structured piece of work that assigns a novel function to the tryptophan repeat domain of Nanog.

Major

- The authors demonstrate the Nanog-Sox2 interaction exclusively using overexpression of tagged constructs. It is also surprising that this interaction between highly expressed partners has not been picked up by previous studies (e.g. Wang et al. PMID 17093407). The authors should therefore also show interaction of the endogenous proteins in ES cells (that are expressed at physiological levels), using anti-Nanog and anti-Sox2 antibodies for Co-IPs.

- The functional analysis for testing the relevance of the interaction (Fig. 7B) seems a bit sparse. Are the differences to 3Y>A statistically significant? What are the effects of 3Y>A expression in the longer term? What are the consequences of the modification on downstream gene expression? Are

known targets co-bound by Nanog and Sox2 affected in their expression levels after several passages? I think the functional aspects should be explored a bit further.

Other

- There seems to be a slight discrepancy between the data in Figs. 5 and 6. From Fig. 6, the authors conclude that the third repeat in Sox2 was important for the interaction but in Fig. 5, the 1-218 and particularly the 1-226 constructs (without repeat 3) perform just as well as the control. Please explain / discuss.
- The methods section, in particular information explaining genetic modifications, is in part hard to follow. For example, I did not understand why there is no endogenous Nanog protein in the right lane of Fig. 1A. In general, methods key to the understanding of the manuscript should be better explained (e.g. briefly mention the main characteristics of cell lines like RCNbetaH-B(t) rather than merely referring to published work) - also applies to preparation of nuclear extracts and proteome analysis. How is "Mascot" defined?
- The discussion seems a bit lengthy. In particular, the authors devote several pages to discussing putative interactors that they didn't deal with in the follow-up experiments. Instead, I would like to learn a bit more about what the authors e.g. think of the fact that Nanog's tryptophan repeat domain apparently also interacts with other TFs such as Nac1 and Nanog itself (Ma et al. PMID 19366700 Mullin et al. 18290762). How do we have to picture a scenario that several proteins interact with Nanog via the same (small) stretch of amino acids?
- Fig. 3C: What is the meaning of showing the Sox2 domain structure?

Referee #3:

The manuscript by Chambers and colleagues is very solid and (i) describes an extensive protein interactome of the pluripotency factor Nanog in embryonic stem cells (extending prior observations); (ii) defines the interaction between Nanog and another key transcription factor, Sox2, using a structure-function approach, and (iii) demonstrates that the Nanog-interacting region of Sox2 is important for the maintenance of embryonic stem cell self-renewal. Together these findings provide interesting additional insight into the activity of Nanog.

I have only a few concerns regarding the data presented in the manuscript: 1. The inclusion of an input lane in the pull-down experiments appears somewhat random and the ratio of IP to input shown is not clear. 2. The expression levels of different Sox2 forms in the functional experiment in Figure 7 should be shown to rule out that the subtle phenotype of the 3Y to A mutant is not simply explained by different expression levels.

Overall, the effect of the 3YtoA Sox2 mutation appears to be much stronger on binding than on ES cell self-renewal. It remains unclear whether the in vivo effect is truly due to loss of the Sox2-Nanog interaction. Similarly, it would be nice if the author could add some more data regarding the importance of this interaction - is Nanog and Sox2 binding to DNA synergistic, does this interaction alter transcriptional output? Thus, I am looking for some more experiments that provide additional insights into the role of this interaction. I am also missing some discussion on whether dimerization of Nanog (which appears to require the Tryptophan stretch) would affect the interaction with Sox2, ie does Sox2 interact with monomeric or dimeric Nanog?

1st Revision - authors' response

09 June 2013

Referee #1:

The paper by Gagliardi et al, "A direct physical interaction between Nanog and Sox2..." provides a comprehensive, proteomic view of partner proteins that interact with Nanog in mouse embryonic stem cells. One of the interacting partners revealed is Sox2. Given that Sox2 and Nanog are both important regulators of pluripotency, the authors went on to biochemically detail the physical

interaction between these two transcription factors. They also show that the pluripotency function of Sox2 is compromised by loss of Nanog association.

This work is original and insightful as it shows the importance of combinatorial interactions between key transcription factors to execute an important biological function. Overall the experiments were performed with care, the data presented clearly, and the manuscript well written. However, few points should be addressed:

We are pleased to note that this reviewer considers our work “comprehensive” and “original and insightful as it shows the importance of combinatorial interactions between key transcription factors to execute an important biological function”.

1. The proteomic analyses were done with high rigor, as previously advanced by the group, to assess other interactomes in ES cells. This paper provides too little detail to the reader without going to referenced papers. At least a brief description of the method is warranted here, particularly noting the use of DNase that precludes indirect associations through co-binding to DNA.

We have extended the “Protein purification” section in the Materials and Methods to address this omission.

2. The F-Nanog cell line used should be better characterized with respect to retaining pluripotency traits-marker expression or differentiation potential should be provided to confirm that the engineered cells are still pluripotent.

We have added data (Fig. 1B) showing appropriate expression levels of Oct4, Sox2 and Rex1 mRNAs in F-Nanog cells.

3. Table I, II: "ZIC3" should be "Zic3"

This error has been corrected.

4. Figure 1D: the "C F-Nanog" lane labels as are a bit confusing due to spacing.

We have re-organised the lettering on this Figure panel so that it is at a 45 degree angle, which we think should resolve any ambiguities.

5. Figure 2: It was difficult to quickly locate Sox2 in the diagram. Perhaps, given the importance of this factor in the paper, it could be color or bold highlighted

As suggested, we have changed the colour of the Sox2 circle to make it stand out more readily.

6. Figure 3C: The Sox2 schematic, in multi-colors, should have the domains labeled, consistent with Figure 4.

This oversight has been corrected.

7. Figure 7B legend: please indicate the time point at which the colonies were scored.

This information (7 days) has now been added to the legend.

8. The discussion section could be sharpened up a bit; currently a bit loose and wordy.

We have altered the discussion to include discussion of our new data but have borne this criticism in mind and have tightened up the discussion to be more to the point.

Referee #2:

In this well-written manuscript, Gagliardi et al. extend the protein interactome of a key self-renewal factor in mouse ES cells, Nanog. They demonstrate for the first time direct interaction with another core transcription factor, Sox2, and, in a series of co-immunoprecipitation experiments, reveal the motifs on both Nanog and Sox2 mediating this interaction. This is clearly structured piece of work that assigns a novel function to the tryptophan repeat domain of Nanog.

We are pleased that this reviewer considers the manuscript “well-written” and that this is a “clearly structured piece of work” that “demonstrate for the first time direct interaction with another core transcription factor, Sox2”.

Major

- The authors demonstrate the Nanog-Sox2 interaction exclusively using overexpression of tagged constructs. It is also surprising that this interaction between highly expressed partners has not been picked up by previous studies (e.g. Wang et al. PMID 17093407). The authors should therefore also show interaction of the endogenous proteins in ES cells (that are expressed at physiological levels), using anti-Nanog and anti-Sox2 antibodies for Co-IPs.

We have added data from wild-type E14Tg2a ES cells (Fig3A) showing:

(1) that immunoprecipitation of endogenous Sox2 co-precipitates Nanog and

(2) that immunoprecipitation of endogenous Nanog co-precipitates Sox2.

These experiments, performed using wild-type ES cells, demonstrate that the Nanog-Sox2 interaction occurs at physiological levels.

- The functional analysis for testing the relevance of the interaction (Fig. 7B) seems a bit sparse. Are the differences to 3Y>A statistically significant? What are the effects of 3Y>A expression in the longer term? What are the consequences of the modification on downstream gene expression? Are known targets co-bound by Nanog and Sox2 affected in their expression levels after several passages? I think the functional aspects should be explored a bit further.

The difference in the functional assays shown (now in Fig 8) are statistically significant (t-test; 2 tails, $p=1.4 \times 10^{-5}$). In the longer term, ES cells that only express the Sox2YYY>A protein differentiate and cannot be maintained. However, the population is contaminated by a small proportion of cells that do not excise the transgene and that overgrow the cultures. This is a problem we hope to solve in future studies.

We have added additional functional data by purifying the Nanog-Sox2 complex from E. coli and determining the DNA motif that the complex binds. Interestingly, this is a composite in which we can recognise a sequence similar to the Sox2 element and a sequence resembling the core homeodomain binding nucleotides. There are a couple of notable features here. The elements are oriented in a specific way with respect to one another, are separated by 2bp and the core homeodomain binding motif differs from that determined by ourselves (this work) and others (Mitsui et al., PMID: 12787504). This suggests that interaction with Sox2 alters the DNA binding specificity of Nanog. The Nanog-Sox2 motif is closely related to one recently described by de novo discovery analysis (Hutchins et al. 2013 Stem Cells 31, 269-281). Using a composite motif we show the existence of this motif in many of the overlapping Nanog and Sox2 peaks from 3 independent ChIP-Seq studies. We then show that some genes that possess this motif are sensitive to the loss of Nanog-Sox2 interaction because the expression of the associated gene is disrupted in Sox2YYY>AAA.

Other

- There seems to be a slight discrepancy between the data in Figs. 5 and 6. From Fig. 6, the authors conclude that the third repeat in Sox2 was important for the interaction but in Fig. 5, the 1-218 and particularly the 1-226 constructs (without repeat 3) perform just as well as the control. Please explain / discuss.

We think this is a misunderstanding, since Fig. 5 shows that the first repeat rather than the third repeat is important for the interaction. Looking at the text we think that the experiment can be described more explicitly, so we have added the following text for clarification:

“additional truncations were made after residues 218 and 226, which truncate Sox2 after repeat 1 or 2 respectively. This indicates that repeat 1 is sufficient for interaction with Nanog but that together repeats 1 and 2 interact with Nanog with an efficiency approaching that of wild-type Sox2 (Fig. 5B). To examine the sequences required on Sox2 in more detail,”

- The methods section, in particular information explaining genetic modifications, is in part hard to follow. For example, I did not understand why in there is no endogenous Nanog protein in the right lane of Fig. 1A. In general, methods key to the understanding of the manuscript should be better explained (e.g. briefly mention the main characteristics of cell lines like RCNbetaH-B(t) rather than merely referring to published work) - also applies to preparation of nuclear extracts and proteome analysis. How is "Mascot" defined?

We have rewritten the methods section to be more explicit and defined “Mascot”. We have added a sentence to the legend of Figure 1, referring to the autorepression of Nanog (Navarro et al., EMBO J. 2012) that explains the lack of a wild-type band in Fig 1A (right panel). We have also added more detail to the description of the RCNbetaH-B(t) cells and to the protein purification section and simplified the description of the Sox2CKO line to clarify what was done.

- The discussion seems a bit lengthy. In particular, the authors devote several pages to discussing putative interactors that they didn't deal with in the follow-up experiments. Instead, I would like to learn a bit more about what the authors e.g. think of the fact that Nanog's tryptophan repeat domain apparently also interacts with other TFs such as Nac1 and Nanog itself (Ma et al. PMID 19366700 Mullin et al. 18290762). How do we have to picture a scenario that several proteins interact with Nanog via the same (small) stretch of amino acids?

We have re-written the Discussion to address the referee's criticism. Specifically, we have shortened the discussion of putative interactions that we do not follow up and we have speculated about the ways in which interacting partners may interact with the WR repeat.

- Fig. 3C: What is the meaning of showing the Sox2 domain structure?

Referee 1 also raised an issue with this panel and we have altered the Sox2 diagram to address the criticism.

Referee #3:

The manuscript by Chambers and colleagues is very solid and (i) describes an extensive protein interactome of the pluripotency factor Nanog in embryonic stem cells (extending prior observations); (ii) defines the interaction between Nanog and another key transcription factor, Sox2, using a structure-function approach, and (iii) demonstrates that the Nanog-interacting region of Sox2 is important for the maintenance of embryonic stem cell self-renewal. Together these findings provide interesting additional insight into the activity of Nanog.

We are pleased that this reviewer considers our manuscript "very solid" and that it provides "interesting additional insight into the activity of Nanog".

I have only a few concerns regarding the data presented in the manuscript: 1. The inclusion of an input lane in the pull-down experiments appears somewhat random and the ratio of IP to input shown is not clear. 2. The expression levels of different Sox2 forms in the functional experiment in Figure 7 should be shown to rule out that the subtle phenotype of the 3Y to A mutant is not simply explained by different expression levels.

Point 1. The number of lanes on the gel rigs that we used in our lab did not always allow inclusion of input next to all IP lanes on a single gel. For immunoprecipitations we always load an equivalent of 1% of the lysate volume used for immunoprecipitation in the "Input" lanes; this information has been added to the relevant legends.

Point 2. We have included immunoblots showing the level of expression of Sox2 (Fig. 8D). This indicates that the reduced self-renewal seen in Sox2YYY>A cells is not due to decreased transgene expression.

Overall, the effect of the 3YtoA Sox2 mutation appears to be much stronger on binding than on ES cell self-renewal. It remains unclear whether the in vivo effect is truly due to loss of the Sox2-Nanog interaction. Similarly, it would be nice if the author could add some more data regarding the importance of this interaction - is Nanog and Sox2 binding to DNA synergistic, does this interaction alter transcriptional output? Thus, I am looking for some more experiments that provide additional insights into the role of this interaction. I am also missing some discussion on whether dimerization of Nanog (which appears to require the Tryptophan stretch) would affect the interaction with Sox2, ie does Sox2 interact with monomeric or dimeric Nanog?

We agree that the effect of the Sox2YYY>AAA mutation appears to be much stronger on binding of Nanog than on ES cell self-renewal. However, it is important to note that loss of Nanog does not obligatorily result in loss of self-renewal and in fact dosage effects of Nanog on ES cell self-renewal have been documented (Chambers et al., PMID: 18097409; Hatano et al., PMID: 15582778). It is our view that a 50% decrease in the number of colonies formed by a Sox2 mutant in which 3 tyrosine residues have been mutated to alanine and in which Nanog interaction is abrogated is quite impressive, particularly since we are able to fully restore the biological function by replacing the mutated alanine residues with phenylalanine.

With respect to the reviewers remaining points, we have added SELEX data on the interaction of the Sox2-Nanog complex with DNA, examined the occurrence of the Sox2-Nanog complex in ChIP datasets and analysed the expression of genes associated with the Sox2-Nanog complex in cells in which the Sox2-Nanog interaction is intact or abrogated. We have also added to our discussion, our

speculation on whether the Sox2-Nanog interaction occurs via monomeric or dimeric Nanog, a point that we consider to be important in our future studies, but one that will require further careful analysis that we consider is outwith the scope of our present manuscript.

Additional Correspondence

12 June 2013

Thank you very much for the revised study that was assessed by one of the original referees.

Before formal acceptance though, we were wondering whether inclusion of a table that integrates/highlights particularly known pluripotency-related genes that carry the dual Sox2/Nanog motif (Fig7) might improve general comprehension of you data?

Further, please notice that The EMBO Journal encourages the publication of source data, particularly for electrophoretic gels and blots, with the aim of making primary data more accessible and transparent to the reader. This entails presentation of uncropped/unprocessed scans for the KEY data of published work. We would be grateful for one PDF-file per figure combining this information. These will be linked online as supplementary "Source Data" files. Please do let me know if you have any questions regarding this initiative.

Irrespective of such potential modifications, please allow me to congratulate you to the study. The editorial office will be in touch with necessary paperwork related to official acceptance upon your response to these minor modifications.

I am very much looking forward to efficient proceedings in this matter.

REFEREE REPORT

Referee #3

Chambers and colleagues included additional data in this revision consistent with a Nanog-Sox2 interaction *in vivo*. This is novel and should encourage more work in this area. This work should be published soon without any delay.

Minor point:

- Figure 2 - The authors may want to consider to highlight common interaction partners in a different colour?